# Color construction of multi-colored carbon fibers using glucose

Sijie Zhou[1,2], Chunhua Zhang[1], Zhuan Fu[1,3], Qimeng Zhu[1], Zhaozixuan Zhou[1], Junyao Gong[1], Na Zhu[1], Xiaofeng Wang[1], Xinjie Wei[1], Liangjun Xia [1]✉ & Weilin Xu [1]✉

Carbon fibers (CFs) have attracted attention in the automotive, aviation, and aerospace industries. However, the coloration of CFs is challenging due to their brittleness, inertness, complexity, and time/energy-intensive processes. Herein, inspired by the naturally grown protrusive nanostructures on the green central surface of peacock back feathers, we report an in-situ self-growing strategy for developing carbon spheres (CSs) on the CFs surface to achieve color tuning. This is achieved via the dynamic growth of CSs using glucose as the feeding material. Combined with the coloration process, the interaction between CSs and CFs promotes stable interfacial forces in integrated molding. This strategy allows the coloring system to continuously vary its color in a designated manner, thereby, endowing it with satisfactory mechanical robustness, acid durability, and light fastness. We anticipate this developed approach can be potentially competitive in the color construction of CFs with multi-colors due to its low-cost manufacturing.

Carbon fibers (CFs) have become very popular with the rapid expansion of CF-based materials, derived from fibrous carbon-based materials with a graphite crystal structure[1]. Furthermore, owing to their superior mechanical properties, excellent electrical conductivity, remarkable thermal stability, high corrosion resistance, and good friction resistance, they are widely used in the automotive, aviation, aerospace, national defense, and military industries[2]. However, the available CF color is limited to black, which cannot satisfy the diversified aesthetic requirements of vivid worlds interspersed with various colors. Thus, owing to the high crystallinity and inadequate chemical affinity of CF, a coloration strategy that was fundamentally different from the conventional dyes, was devised to develop the color of CF by physically assembling spheres or chemically growing periodic nanostructures[3,4], attributable to the structural color that evolved CF from black to the devisable color.

Structural color, arising from the interaction of visible light with materials containing specific micro/nano-structures, is durable since it originates from extensively investigating physical phenomena, such as reflection, refraction, and polarization, which are widespread in natural organisms, including birds, fishes, insects, and flowers[5,6]. Nature provides vast inspiration for designing intense colors based on the mechanism of structural coloration, thereby reducing dependence on pigments. Owing to detailed investigations on the various optical phenomena, these natural counterparts have been used to fabricate artificial structural color materials. Moreover, achievements have been made in the construction of artificial structural colors derived from metals, inorganic compounds, natural biopolymers, and synthetic organics, thus demonstrating versatility in the utilization of materials and manufacturing structures, such as $SiO_2$[7], single-crystal $Cu_2O$[8], monodisperse ZIF-8[9], $SnO_2$ inverse opals[10], cellulosic films[11], polystyrene beads[12], binary block copolymer[13], and poly[styrene-co-(butyl acrylate)-co-(acrylic acid)][14]. Structural colors based on inorganic and synthetic polymer materials have emerged as potential choices for mass production owing to their widely designed physicochemical features. While natural polymers have attracted considerable attention, reestablishing of biopolymers into structurally colored materials has been identified as a primary consideration, due to the superiorities of abundance, renewability, biocompatibility, biodegradability, and

[1]State Key Laboratory of New Textile Materials and Advanced Processing Technologies, Wuhan Textile University, Wuhan 430200, China. [2]College of Textiles, Donghua University, Shanghai 201620, China. [3]College of Textile Science and Engineering, Zhejiang Sci-Tech University, Hangzhou 310018, China. ✉e-mail: liangjun_xia@wtu.edu.cn; weilin_xu@wtu.edu.cn

environmental friendliness[15]. Natural polysaccharides are macromolecules with the repetitive structural unit of monosaccharides, which are critical components of plant cells that contribute to constructing structural colors. The strategy of artificial structural color derived from these natural polymers with tailored structural and optical features has reduced the burden on the environment and the energy consumption of fabrication.

Owing to the distinct structure of natural polymers, creating artificial structural colors directly from biopolymers could aid in understanding the functioning of natural creatures and investigating structural coloration mechanisms[16]. Consequently, the ability to reconstruct structural colors by modifying natural polymers or extracting biopolymers from natural materials has improved. Furthermore, to achieve rapid fabrication and color diversity, various manufacturing techniques have been employed, including spray synthesis[7], casting[17], dip-coating[18], spin coating and UV polymerization[19], infiltration-driven nonequilibrium assembly[20], electrophoretic deposition[21], atomic layer deposition[22], and optical lithography[23]. A common characteristic of these methods is that they depend on the evaporation or volatility of the solvent to provide adequate time for particles to assemble.

For structural colors, the reflected light depends on the nanostructure and refractive index of the material, thus confirming the potential of photonic crystals (PCs) attributed to the highly ordered and periodic structures of short- and long-range orders[24,25]. In this fabrication concept, disordered structures have long been considered natural adversaries of optical systems, often causing them to deviate from the ideal design. Minimizing residual disorder is essential in the formation of PCs. The disorder is always present and never fully eliminated in the artificial structural color[26]. Nevertheless, inspired by disordered structures derived from nature, structural colors were acquired by a disordered arrangement[27], whereby light was reflected or transmitted at specific wavelengths by design[16,28]. With the popularity of artificial random media with novel properties, conspicuous physical phenomena arise from disordered systems; these include selectively dissolved polystyrene spheres (PS) in the ordered structure of poly(methyl methacrylate) (PMMA)-PS spheres[29], random aggregation of monodisperse colloidal PS spheres[30], a geometric disorder of plasmonic nanostructures[31], amorphous photonic structures of mixed mesoporous hollow $TiO_2$ and $SiO_2$[18], and selective etching in a binary system of PS-PMMA spheres[26]. Notably, disordered structures that strongly affect light transport have been avoided during structural color formation due to their low spectral selectivity, which generates low color saturation. Fourier transform was used to elucidate this disordered structure phenomenon, in which the zero position of the motif Fourier transform was traced to the right side of the prominent peak for the structure factor in reciprocal space[32]. Compared with the reflection of one selective color produced by a steep transition due to a smooth spectral transition from weak to strong reflection, reflected light consists of a mixture of colors, that causes low structural color saturation. Special black materials, such as graphene nanosheets with graphene quantum dots[33], polydopamine shell layers[34], eumelanin nanoparticles[35], and cuttlefish ink particles[36], designed to absorb visible light and reduce incoherent light scattering to improve color saturation, were investigated to overcome this problem.

In this work, inspired by the naturally grown protrusive nanostructures on the green central surface of peacock back feathers, a randomly disordered self-growing structure of carbon spheres (CSs) was constructed via dynamic in-situ growth to achieve multi-colored CFs. This study focused on the intrinsic disorder from the formation process of one-step hydrothermal carbonization (HTC) using glucose as feeding material, which was further employed to design multi-color characteristics by controlling the glucose concentration as the factor. The developed method ensures the stability of the interacting forces between CSs and CF, thereby preventing them from sticking and aggregating on CF. Moreover, the hydrothermal reaction process of glucose endows the strong interface between CSs and CF, thus enhancing the color stability of CFs, even experiencing the rigorous examinations of mechanical friction, acid immersion, and ultraviolet–visible light irradiation. The findings in this work may provide an alternative approach for the coloration of CFs using a low-cost way, which may further broaden the potential applications of glucose in the color construction of high-performance fibers to meet industrial requirements.

## Results

### Design and preparation of colored CF fabrics

A striking structural color has been created by living organisms on the central area of the peacock back feather, exhibiting the unique characteristic of angle-independent property (Fig. 1a). Meanwhile, as shown in Fig. 1b and Supplementary Fig. 1, on the barbule surface for the central area of the feather, the protrusive nanostructures are formed during the natural growth of the feather. Inspired by this protrusive nanostructure, we proposed a strategy for the random in-situ growth of CSs on the CF fabric only using glucose as the feeding material (Fig. 1c, d). The CSs distributed and grew randomly by interacting with other CSs and CFs to form a structural color on the CF fabric, and the corresponding color tuning was designed by controlling glucose concentration during the one-step hydrothermal carbonization (HTC). As displayed in Fig. 1e, the photographs show the concentration-dependent colored CF fabrics using the as-prepared method. The colors of 4CC, 7CC, 10CC, 13CC, and 17CC are shown in blue, yellow, orange-red, purple, and green, respectively. Supplementary Fig. 2 shows photographs of 5CC, 6CC, 8CC, 9CC, 11CC, 12CC, 14CC, 15CC, and 16CC, derived from the various glucose concentrations. Compared with the raw CF fabric, the colored CF fabrics were aesthetically pleasing in extensive colors, essentially overcoming a dark and monotonous appearance. By precisely tuning the concentration of the original glucose solution, multi-colored CF fabrics were achieved, demonstrating the superior adaptability of the in-situ growth of CSs in the color construction of high-performance CFs.

To further visualize the color gamut distribution, the colors of various samples were investigated, whose corresponding CIE chromaticity coordinates widely scattered in orange-red, yellow, green, purple, and blue with each other in Fig. 1f. Moreover, the characteristic wavelengths of colored CF fabrics in the reflectance spectra were recorded at 400, 430, 470, 540, and 570 nm, respectively, demonstrating for blue (400 nm), yellow (430 nm), orange-red (470 nm), purple (540 nm), and green (570 nm), according to the peaks (4CC and 17CC) and dips (7CC, 10CC, and 13CC) in Fig. 1g. As shown in Supplementary Fig. 3, the K/S value at the characteristic wavelength was used to assess the color stability and the characteristic peaks in K/S spectra correspond to the maximum K/S values at 700, 430, 470, 540, and 700 nm, matching with the dips of reflectance spectra. The diameter distribution and representative average diameter of CSs were estimated using the Gaussian function based on the diameter measurement of CSs on the colored CFs (Supplementary Fig. 4). The CSs with average diameters of 212.0, 258.7, 282.9, 308.6, and 350.0 nm on the CFs contributed to the blue, yellow, orange-red, purple, and green structural colors, respectively. The relationship curve between the characteristic wavelengths and CSs diameters for various colors is displayed in Fig. 1h, further demonstrating that the average diameter of CSs on the CFs can tune the diversity of colors by controlling the concentration of the glucose solution. The colors were distributed around the white point in a chromaticity diagram with changes in the average diameter of CSs, which is mainly because the diameter determines the color hue[30]. Explanations of the correlation between the diameter and the hue for structural color have been proposed. As the size-dependent structural color, the diameter can tune the scattering properties of materials to control the colors[18,30,31,37]. The structural color of this disordered structure was determined by the scattering properties of the CSs, whose Mie scattering of individual

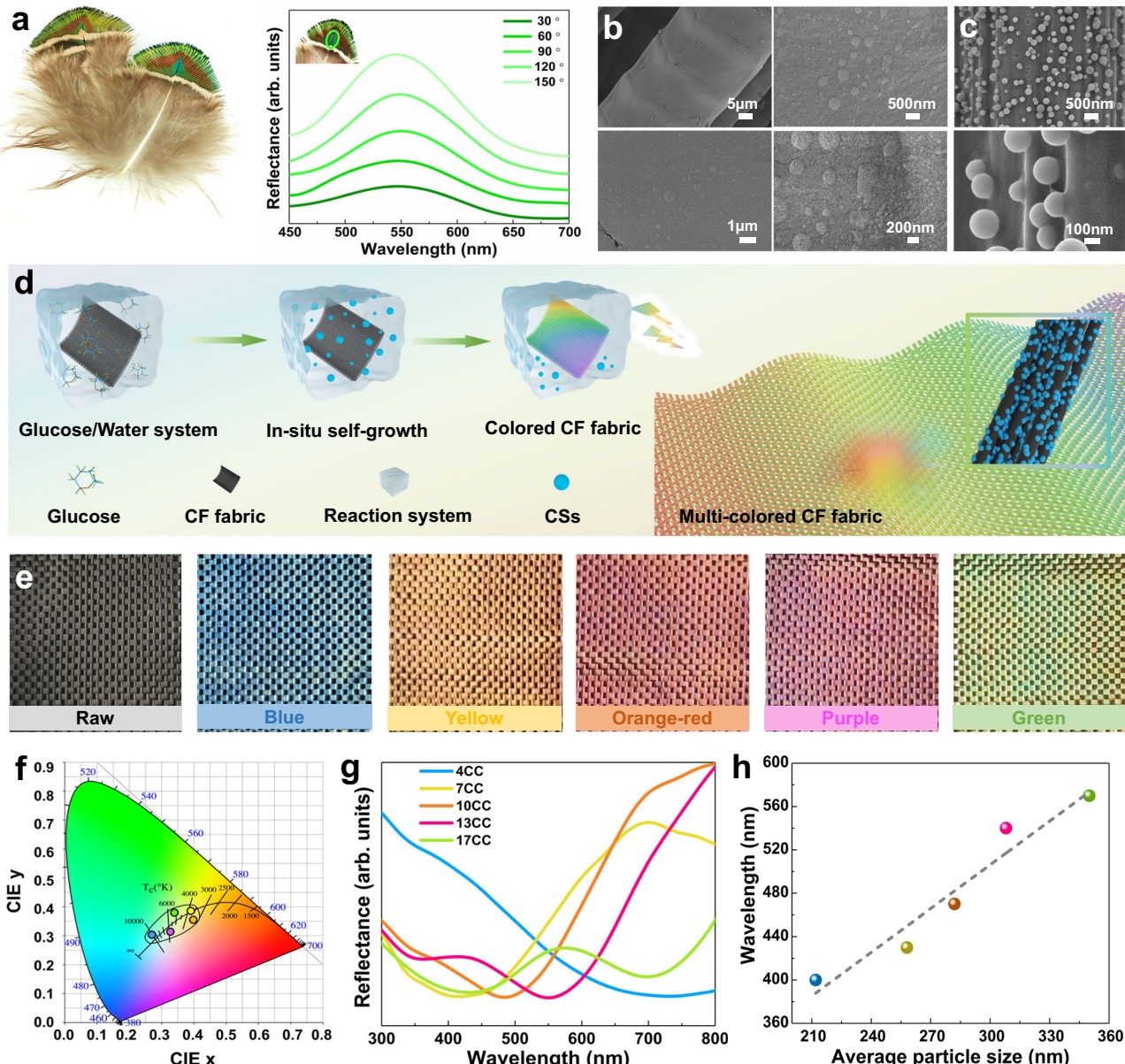

**Fig. 1 | Scheme for design and preparation of multi-colored carbon fiber (CF) fabrics. a** Photograph of the peacock back feather and the angle-resolved reflectance spectroscopy of the central area (green) for the peacock back feather. **b** SEM images of the barbule surface at the central area of the peacock back feather. **c** SEM images of in-situ grown carbon spheres (CSs) on the CF surface by one-step hydrothermal carbonization (HTC). **d** Fabrication process of the various structural colors for CF fabrics and the conceptual illustration of in-situ self-growth of CSs on CFs. **e** Photographs of raw CF fabric and structurally colored CF fabrics. **f** CIE diagram showing the color shift of 4CC, 7CC, 10CC, 13CC, and 17CC for the diameter change with respective x–y coordinates of (0.2716, 0.3019), (0.3909, 0.3814), (0.3955, 0.3600), (0.3276, 0.3165), and (0.3436, 0.3732), respectively. **g** Reflection spectra of 4CC (blue), 7CC (yellow), 10CC (orange-red), 13CC (purple), and 17CC (green). **h** Relationship of characteristic wavelengths as a function for the average CSs diameter.

spheres may completely overwhelm the optical effect of diffraction resonances[38]. Consistent with the disordered structure color, the interacting light depends on the scattering in all directions assigned to the major Mie scattering[38,39].

## Formation of structural color by in-situ self-growth of CSs

The CSs structure on the colored CFs surface was revealed by SEM. As shown in Fig. 2a and Supplementary Fig. 5, the CSs distributed on the surface of CFs exhibit frontal spherical morphologies and lateral hemispherical structures. The stability was evidenced by the high-resolution SEM images of the shape of CSs attaching to the CFs surface. A side view of the colored CFs exhibited interfacial adhesion between CSs and CFs from the various areas of colored CF fabric, which is possibly due to the strong interactions of this lateral hemispherical

structure (Supplementary Fig. 6), contributing to the stability of the structural color. As shown in Fig. 2b and Supplementary Fig. 7, compared with the XRD patterns of CSs, the crystal structure of glucose disappeared, with increasing glucose concentration, no significant changes were observed in the crystal structure of the CSs. The broad characteristic peaks at $2\theta = 20.6°$ suggest a disordered structure and low degree of graphitization for the CSs[40]. Figure 2c presents recognizable peaks of the CSs FTIR spectra for various glucose concentrations, indicating the presence of aromatic units, furan ring, and carbonyl groups in saturated aliphatic ketones (Supplementary Note 1). NMR spectrum is used to mainly investigate the aromatization of the polyfuranic compounds, as displayed in Fig. 2d. The peak assignments could be divided into three main regions (Supplementary Note 2), which were identified as aliphatic carbons, aromatic or

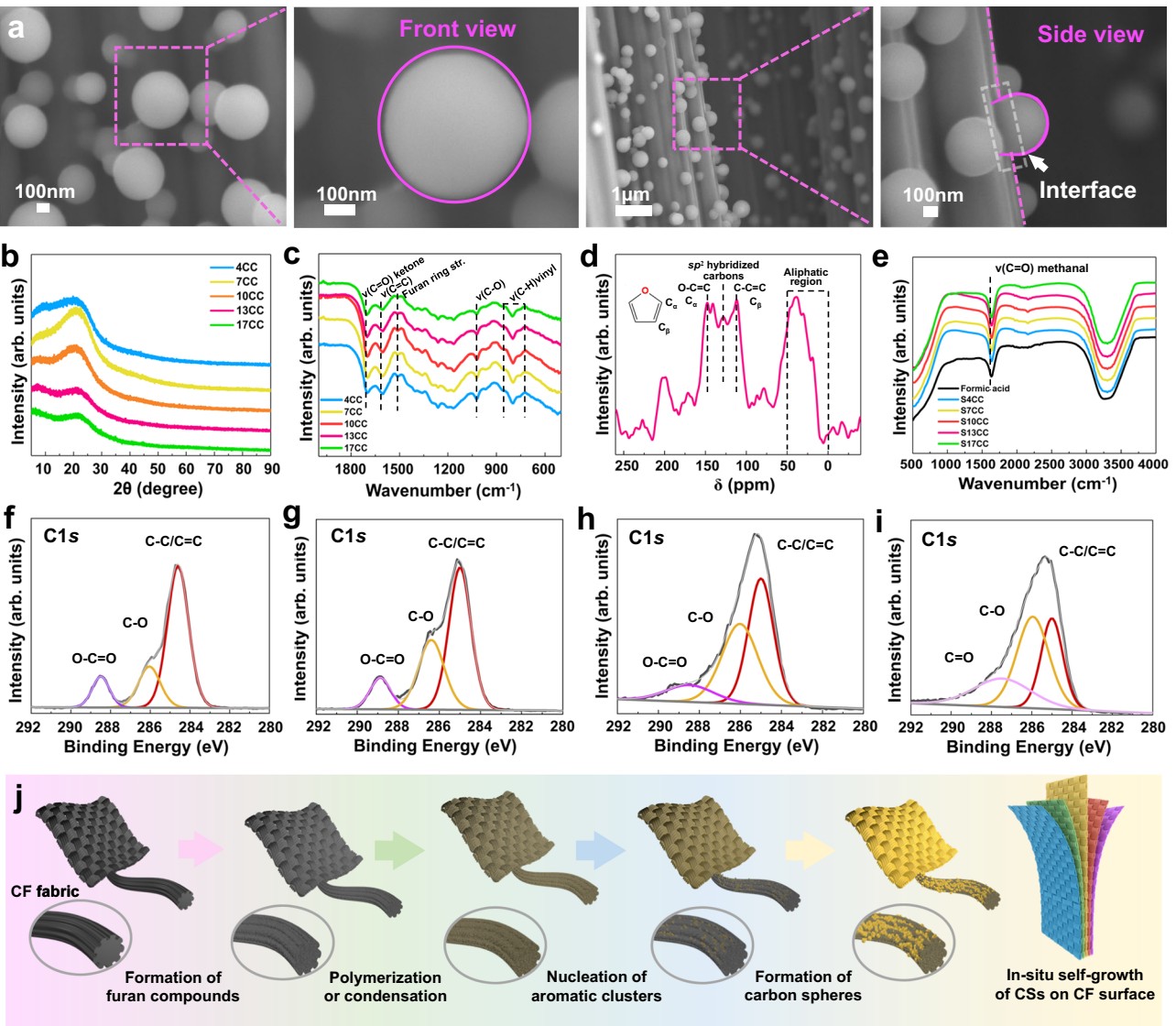

**Fig. 2 | Formation of structural color by in-situ self-growth of CSs. a** SEM images of frontal and lateral morphologies of CSs on the colored CFs. **b** XRD curves of solids after the HTC process for 4CC, 7CC, 10CC, 13CC, and 17CC, respectively. **c** FTIR spectra of solids after the HTC process with various concentrations of glucose (XCC: X g 70 mL$^{-1}$). **d** Solid-state carbon-13 MAS NMR spectra of 13CC. **e** Supernatant solution separated from residue after the HTC process with various concentrations of glucose (SXCC: supernatant solution separated from residue of HTC process for X g 70 mL$^{-1}$). C1s spectra after hydrothermal reaction of glucose from various stages (**f**) 30 min, (**g**) 60 min, (**h**) 90 min, and (**i**) 120 min at 250 °C. **j** Scheme for the possible formation process of CSs on the CF surface.

unsaturated carbons, and carbonyl groups related to the signals at $\delta < 50$ ppm, 90 ppm $< \delta < 150$ ppm, and $\delta > 150$ ppm, respectively[41]. The two intense signals observed at 140−150 ppm and 110−120 ppm were assigned to the O−C=C and C−C=C of Cα and Cβ for the furan rings, respectively. An apparent signal observed at 125−130 ppm was identified as a symbol for aromatization[42], thus demonstrating furan units and further aromatization in the chemical structure of CSs. Five conspicuous peaks are observed in the CSs mass spectra at 318.4, 334.7, 381.9, 404.2, and 573.0 m/z (Supplementary Fig. 8), which corresponded to a family of oligomers[41]. The supernatant solution separated from the residue after the HTC process was investigated by the FTIR spectrum (Fig. 2e), which revealed two apparent peaks. The signals at 1630 and 3000−3700 cm$^{-1}$ were attributed to C=O and −OH, respectively. Formic acid is a fundamental reaction product of monosaccharides[43], which is consistent with the HTC of glucose promoting the reaction. The TEM images of CSs demonstrate the core-shell structure of CSs obtained in this work (Supplementary Fig. 9). The core-shell structure is proposed to clarify the growth of the carbonaceous materials, which shows that the value of carbon content was 67.17% under the reaction temperature of 250 °C and time of 2 h (Supplementary Table 1), thus producing a carbon-rich CSs[44,45].

As reported by Hu et al.[46], a low-temperature HTC process was conducted up to 250 °C, employing several chemical transformation processes, including dehydration, polymerization, condensation, and aromatization[47]. During the hydrothermal treatment of glucose, 5-hydroxymethylfurfural (HMF), a reaction-driven dehydration product, was formed in the initial stage[48], with formic acid as the degradation product[49]. The acidic conditions promoted the further dehydration of glucose into HMF, which subsequently underwent polymerization-polycondensation reactions contributing to the formation of polyfuranic compounds[41,45]. During this polymerization step, the aromatization of soluble polymers is then produced via intramolecular dehydration with separating from the aqueous solution to form nucleation and growth[42]. The photographs of reaction products and SEM images with the corresponding reaction times are demonstrated in Supplementary Fig. 10, indicating that the CSs were formed gradually on the surface of CF with the hydrothermal reaction of glucose. To further explain the growth process of CSs, *XPS* and NMR were

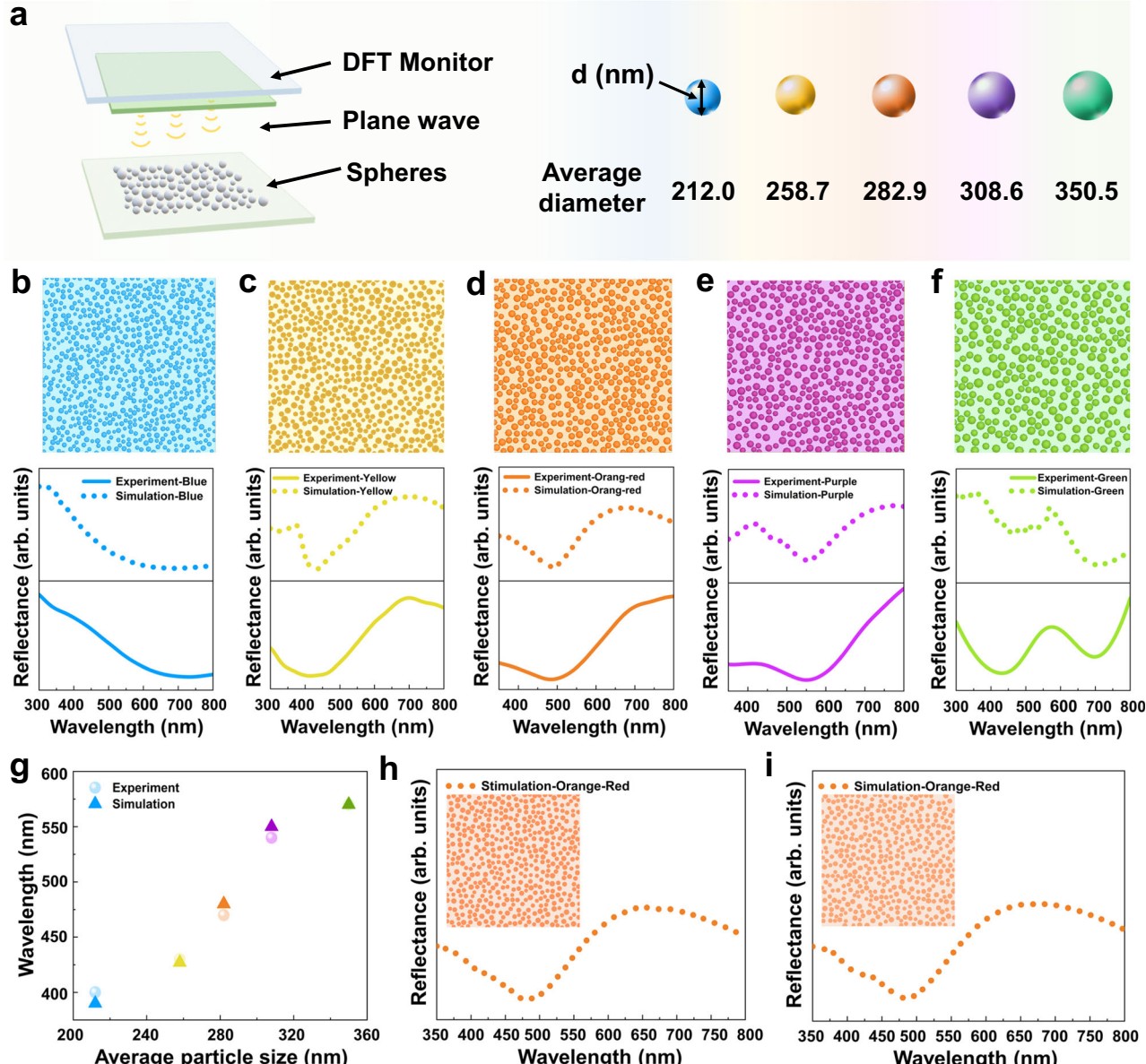

**Fig. 3 | Simulations for various structural colors using FDTD. a** Schematic illustration of FDTD simulations for CSs with different average diameters. **b–f** FDTD-Simulated reflection spectra based on the average diameter of 212.0, 258.7, 282.9, 308.6, and 350.5 nm with a random distribution. **g** Diameter dependence of dip or peak position in reflectance and FDTD simulations reflectance spectra. **h, i** FDTD-simulated reflection spectra of CSs with the average diameter of 282.9 nm at different chaos.

performed to investigate the products at the various stages of the hydrothermal reaction. Before the formation of aggregations on the surface CF, comparing to the raw glucose (Supplementary Fig. 11a), the functional group of the products has changed markedly with the enhancement ratio of C/O and the formation of O–C=O, and C=O bonds (Fig. 2f–i and Supplementary Fig. 11b–e), which is mainly due to the dehydration and decomposition in the process of hydrothermal reaction[50]. Furthermore, in the early stages of the reaction (Supplementary Fig. 12a), the solution [13]C NMR spectrum was characterized by the presence of the two peaks (140–153 and 110–120 ppm), which are assigned to the a polymer-like structure composed of polyfuranic chains domains[42]. With the extension of reaction time, the obvious intensity of the peak at 125–129 ppm demonstrates the aromatization of the polyfuranic compounds (Supplementary Fig. 12b). Compared to the solid [13]C NMR spectra of 90 min (Supplementary Fig. 12c), the relative intensity of the peak at 125–129 ppm enhanced (Supplementary Fig. 12d), demonstrating the increased degree of aromatization. The

furan compounds were formed at the early stage, accompanied by polymerization or condensation of the intermolecular dehydration and fragmented products. The aromatization of soluble polymers was then produced via intramolecular dehydration, which caused the continuous formation of insoluble products promoting nucleation of aromatic clusters and growth. An illustration for the formation process of CSs on the CF fabric is demonstrated in Fig. 2j. The structural colors are constructed in the HTC process of glucose during the formation of CSs. The in-situ self-growth process of the CSs on the CF surface contribute to the constructing the structural color of CF fabric. Compared with conventional methods (Supplementary Table 2), in-situ self-grown CSs by hydrothermal reactions have distinct advantages in constructing the structural color of CF fabric as a low-cost material and a simple method to meet the diversity of CF colors.

As illustrated in Fig. 3, FDTD simulation was utilized to discuss the mechanism of color generation with a random distribution of different CSs diameters. Mie scattering characteristics reflection spectra based

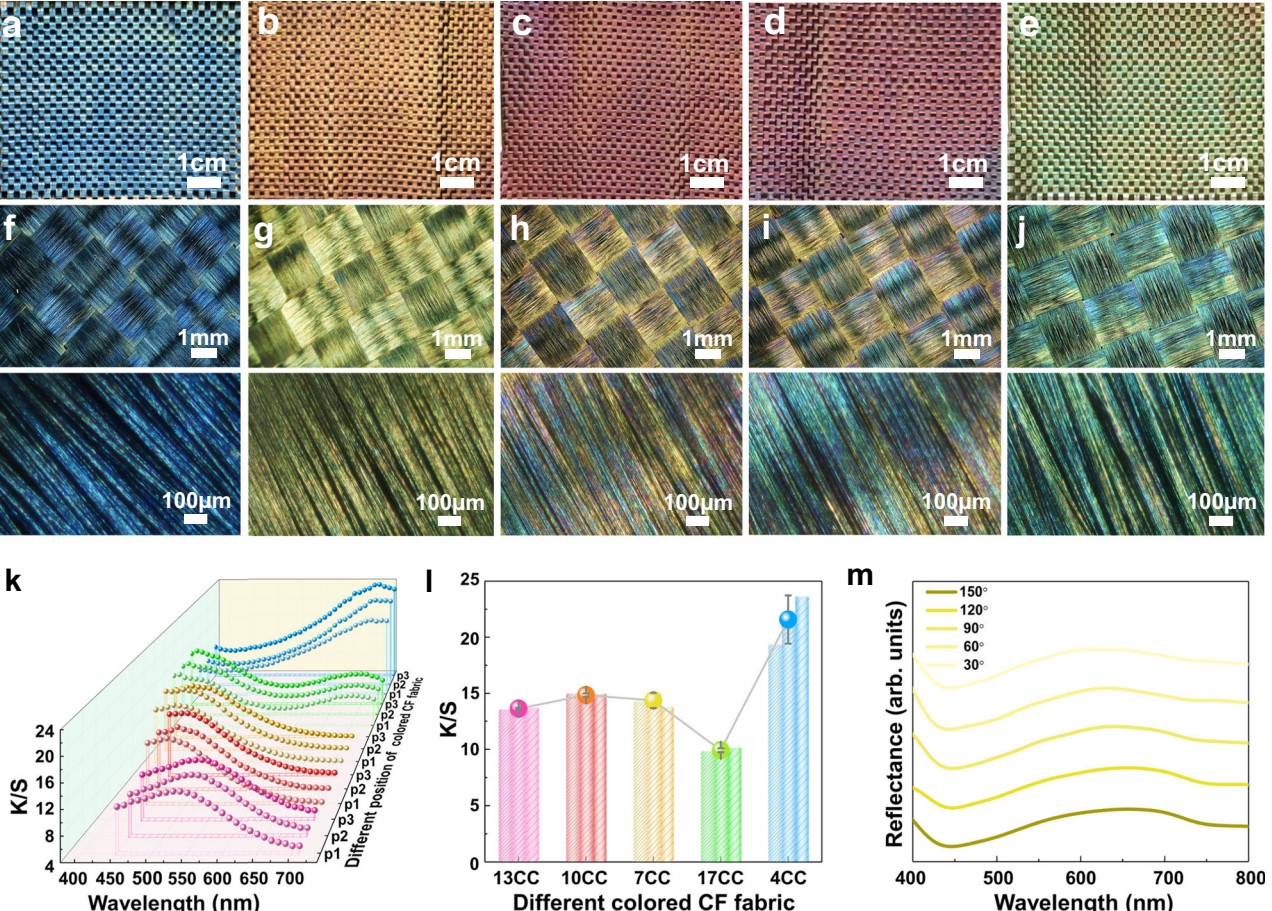

**Fig. 4 | Structural color and optical properties of multi-colored CF fabrics.**
**a–e** Photographs of 4CC (blue), 7CC (yellow), 10CC (orange-red), 13CC (purple), and 17CC (green), respectively. **f–j** Relevant 3D optical microscope images with the 90° angle of incidence (high luminance LED light). **k** K/S spectra at three different positions on 4CC, 7CC, 10CC, 13CC, and 17CC, respectively. **l** Maximum K/S values at characteristic wavelength according to the K/S spectra at three different positions and the average of K/S. Error bars represent standard deviation based on the different positions ($n = 3$). **m** Angle-resolved reflectance spectroscopy of the colored CF fabric (7CC as the representative).

on average diameters of 212.0, 258.7, 282.9, 308.6, and 350.5 nm were simulated to disclose various structural colors (Fig. 3a). Furthermore, FDTD-simulated reflection spectra of 282.9 nm average diameter with various spheres diameter at different chaos were used to investigate the influence of random distribution on color. The refractive index of the CSs is shown in Supplementary Fig. 13, which varies with the wavelengths. To get closer to the random distribution in the experiment, we simulated the reflectance spectra with the random chaos for multi-spheres. As shown in Fig. 3b–f, chaotic spheres within the range of diameter distribution, which represent the CSs on the surface of different colored CF fabrics, thus supporting the possibility of achieving consistent reflection spectra. The simulated reflectance spectra calculated by Mie theory showed a good accordance with the experimental reflectance spectra of the sphere random distribution. More importantly, the color varies with the average diameter of CSs, which was contributed by Mie scattering from the individual nanospheres. Figure 3g shows the characteristic wavelengths of the dip or peak in the stimulated reflectance spectra. The slight mismatch between the simulated and the experimental spectra may be attributed to the inaccurate measurement of the refractive index and the average diameter, which cannot fully representation of sphere distribution[51,52]. Nevertheless, the characteristic wavelengths of dip or peak in the simulated reflectance spectra vary with the average diameter, which is consistent with the dip or peak in the experimental reflectance spectra. Furthermore, based on the Mie scattering, we simulated the characteristics spectra with an average diameter of 282.9 nm, which was

conducted under different chaos. Compared to the simulated reflectance spectrum in Fig. 3d, under the same average diameter, the diverse random distributions of different diameters show the same characteristic wavelength representing the hue of color. The characteristic wavelengths remain unchanged with the different chaos of the spheres, as shown in Fig. 3h, i. Therefore, the simulation indicates that the color generation of the colored CF fabrics accords with the Mie scattering theory, and the color hue is dependent on the average diameter of the CSs on the surface of CFs.

### Optical properties and structural color stability
Figure 4a–e shows the photographs of the major five colored CF fabrics, which display vivid structural colors. The oriented fiber and weaving construction of the CF fabrics contributed to the color presentation. The recorded three-dimensional (3D) optical microscopy images (Fig. 4f–j and Supplementary Fig. 14) correspond to the photographs in Fig. 4a–e and Supplementary Fig. 2. The colors presented in the 3D optical microscopy images of the colored CF fabrics differed from photographs, in which the images exhibited different colors, which may be mainly due to the structural colors consisting of composite colors formed by the combination of two or more primary colors[53]. The color homogeneity of the CF fabric for 4CC (blue), 7CC (yellow), 10CC (orange-red), 13CC (purple), and 17CC (green) was further investigated by measuring the K/S spectra at three different positions. As shown in Fig. 4k, the colored CF fabric displays a relatively uniform distribution of K/S curves at different positions,

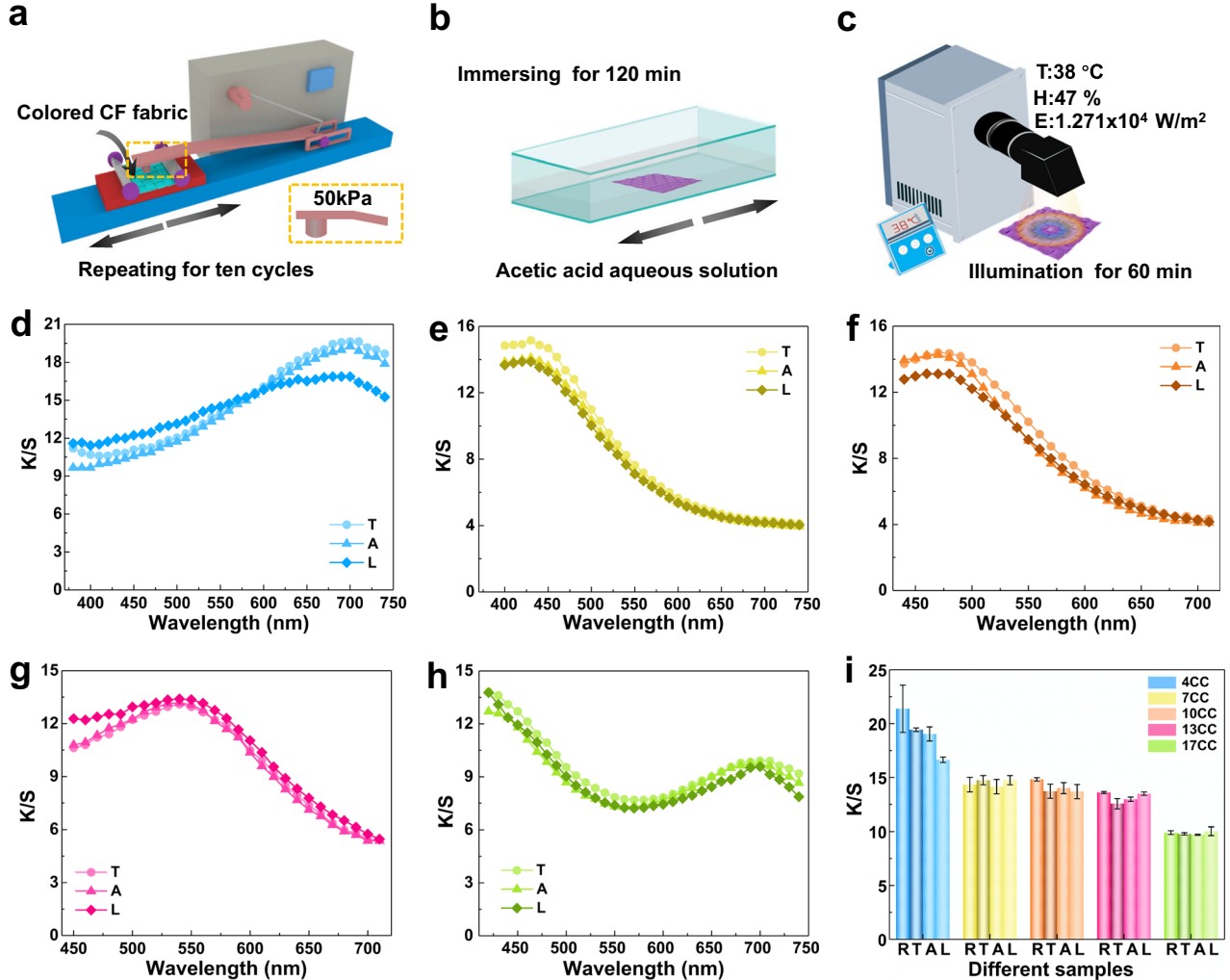

**Fig. 5 | Structural color stability of multi-colored CF fabrics. a** Schematic illustration of the mechanical rubbing test performed by color fastness friction meter with a load pressure of 50 kPa for 10 cycles. **b** Schematic illustration of soaking and washing of acetic acid solution test after mechanical rubbing test at a vibration of 60 times per minute for 120 min. **c** Schematic illustration of the accelerated light-aging test under the strong light using a xenon lamp light source system after mechanical rubbing test and soaking and washing of acetic acid solution (environment temperature, humidity, and light irradiance were set as 38 °C, 47% RH, and 1.271 × 10$^4$ W m$^{-2}$, respectively). **d–h** K/S spectra of the fabricated CF fabrics and (**i**) K/S value at characteristic wavelengths at three different positions after the mechanical rubbing, soaking and washing of acetic acid solution, and accelerated light-aging test for 4CC, 7CC, 10CC, 13CC, and 17CC (T, A, and L were the colored CF fabric after mechanical rubbing test, acid pickling, and accelerated light-aging test, successively). Error bars represent standard deviation based on the different positions (*n* = 3).

essentially illustrating a homogeneous color of CF fabric. As expected, Fig. 4l shows the stable K/S values at characteristic wavelengths, representing the color depth, and demonstrating its consistency in color appearance. To further investigate the directional independence of colored CF fabrics, the angle-resolved reflection spectra were investigated. As shown in Fig. 4m, the colored CFs fabrics exhibit the same reflection peaks when the fixed angle of the reflection source is adjusted from 30° to 150°, which illustrates an obvious angle-independent structural color. Therefore, the developed color construction process may potentially promote the stable color of CFs.

Furthermore, color fastness was evaluated by mechanical stability, acid pickling, and light fastness, which were expressed via K/S spectra and K/S values at characteristic wavelengths. According to ISO 105-X12, the color fastness testing was performed using a color fastness friction meter, as shown in Fig. 5a. After expiring 10 cycles of mechanical rubbing at a load pressure of 50 kPa load pressure, soaking and washing tests were further performed by immersion the fabric in an acetic acid (CH$_3$COOH, 50 wt%) water solution under vibration of 60 times per minute for 120 min (Fig. 5b). Light fastness is a crucial

parameter for practical applications, which was conducted after mechanical rubbing and acid pickling. Under the environmental temperature, humidity, and environmental temperature of 38 °C, 47% RH, and 1.271 × 10$^4$ W m$^{-2}$, the accelerated light-aging was used to monitor the color stability at extremely harsh environmental conditions (Fig. 5c). As shown in Fig. 5d–h, the stable colors of CF fabrics were demonstrated by the no-obvious change of K/S spectra and K/S values at the characteristic wavelengths. To further investigate the color consistency, the average K/S values were calculated from the K/S spectra at three different positions of the colored CF fabric after mechanical rubbing, soaking and washing with the acetic acid solution, and accelerated light-aging. The appreciable stability of the K/S values (Fig. 5i and Supplementary Table 3) at characteristic wavelengths and the color presentation of the photographs (Supplementary Fig. 15) for 4CC, 7CC, 10CC, 13CC, and 17CC demonstrated excellent resistance to mechanical and chemical treatments of the as-prepared CF fabrics. Furthermore, the electrical and mechanical properties of colored CF fabrics were also investigated. Compared with the raw CFs bundle, the surface contact resistance of the colored CFs was enhanced due to the

stable contact between the CSs and the surface of the CFs (Supplementary Fig. 16). As demonstrated in Supplementary Fig. 17, the mechanical performance of the colored CF bundle was enhanced with an evident increase in tensile strength.

The two-dimensional Fourier transform (2D FT) patterns were applied to verify the disordered structure of CSs on CFs. As shown in Supplementary Fig. 18, compared with the 2D FT pattern of the raw CFs, the 2D FT patterns of the structurally colored CFs were all transformed into a bright point at the center without concentric circles. This phenomenon demonstrates the loss of order for the CSs, where the rings in the 2D FT pattern indicate that the correlation extends in space[38]. The different particle sizes act as randomizing spacers, thus creating individual and isolated scatterers in an evident disorder. By introducing disorder into the CF, the typical spectral features of the optical response were based on single particles, where the Mie scattering of individual spheres became the dominant optical effect[38,54]. Using a random ensemble of spheres, single-particle resonances remained visible in the overall optical properties of the material. This random material scatters light, thus leading to structural coloring effects, in which the disordered structure helps to define color, as demonstrated in the 2D FT patterns of the SEM image of the colored CF fabric. Therefore, the application of in-suit self-grown CSs structures on CFs is intrinsically disordered to a certain extent, and the disorder is maximized to obtain an exceptional color appearance.

### Textile weaving using multi-colored CF yarns

The durable carbon sphere structure and strong interactions between CSs and CFs afforded potential possibilities for the weaving process. To further demonstrate the color stability of the samples, we conducted a mechanical rubbing test at a load pressure of 50 kPa for 30 cycles. Consistency of characteristic wavelengths of the reflectance spectra (Supplementary Fig. 19a–e), K/S values (Supplementary Fig. 19f and Supplementary Table 4), and 3D optical microscope images (Supplementary Fig. 20) demonstrated that the colored CF fabrics exhibited satisfactory fastness to the mechanical rubbing of 30 cycles. The multi-colored CF yarns were prepared using the above-developed method to verify their applicability. The one-step hydrothermal reaction was conducted by regulating the glucose construction, contributing to the in-situ self-growth of CSs on the surface of CF yarns (Fig. 6a). The multi-colored CF yarn produced by this method is shown in Fig. 6b. Therefore, the colored CFs were adapted to the varying mechanical requirements of industrial weaving machines. Figure 6c, d shows the schematic illustration for the weaving fabrication of multi-colored CF fabric using the as-prepared multi-colored yarns. Here, we fabricated a plain multi-colored CF fabric on the weaving machine using the as-prepared CF yarns as both the warp and weft yarns (Fig. 6e–g). Although the CF yarns generally experienced severe mechanical rubbing with the heddles of the weaving machine during the weaving process, a piece of $15 \times 15\,cm^2$ plain CF fabric was successfully produced and exhibited satisfactory multi-color (Fig. 6h). Owing to the following main reasons, the developed method may be potentially competitive in the industrial production of colored CFs. Firstly, glucose is used as the only feeding material in the color construction process, which is relatively low-cost and easily obtained. Secondly, glucose concentration was the only factor affecting the color variety under the constant reaction time and temperature. Therefore, we anticipate this developed approach can be potentially applied in the industrial color construction of CFs with multi-colors.

## Discussion

In summary, inspired by the naturally grown protrusive nanostructures on the green central surface of peacock back feathers, we developed a relatively low-cost strategy based on the one-step hydrothermal reaction of glucose for the color construction of structurally colored CFs.

The CSs can be randomly in-situ grown on the surface of CFs, contributing to the color tunning of CFs. The developed approach used low-cost and easily available glucose as the feeding material, and the corresponding color variety was regulated by controlling glucose concentration during hydrothermal carbonization, which can be expected to be industrially produced. Owing to the strong interfacial interaction between CSs and CFs in the self-growth of CSs, the five major colors including blue, yellow, orange-red, purple, and green could be successfully achieved by regulating the distribution and size of CSs in the dynamic equilibrium of the hydrothermal reaction system. The simulation indicates that the color generation of the colored CF fabrics accords with the Mie scattering theory, and the color hue is dependent on the average diameter of the CSs on the surface of CFs. Most importantly, the colored CF fabric shows satisfactory color stability, including withstanding cyclical mechanical rubbing at a load pressure of 50 kPa, acid pickling within 120 min, and accelerated light aging under light irradiance of $1.271 \times 10^4\,W\,m^{-2}$ successively, demonstrating mechanical robustness, acid durability, and ultraviolet-visible light fastness. In the future, we anticipate the developed approach may be a potential choice for the color construction of other high-performance fibers.

## Methods

### Materials

D-(+)-glucose ($C_6H_{12}O_6$, 99.5%) was purchased from Aladdin Co. Ltd. Carbon fiber (CF) fabrics were acquired from Zhongfu Shenying Carbon Fiber Co., Ltd. CF yarns (Type, A-38; Filaments, 06K-D012; AKSACA) was purchased from Dowaksa Ileri Kompozit Malzemeler Sanayi Ltd. Sti.

### Preparation of the color carbon fiber fabric

The colored CF fabric was prepared using a traditional hydrothermal reaction, in which the D-(+)-glucose was used as the carbon source. Firstly, the pretreatment of raw CF fabric was performed in the mixture for 48 h, which consisted of acetone, ethyl alcohol, and deionized water with the volume of 1:1:2 (Acetone ($CH_3COCH_3$, 99.5%), ethyl alcohol ($CH_3CH_2OH$, 99.98%)). Then, the CF fabric was dried at 80 °C. Secondly, the pretreated CF fabric was immersed in the glucose solution, where 7 g of glucose was directly dissolved in 70 mL of deionized water at room temperature. Further, the glucose solution and the sample were transferred into a 100 mL high-temperature reactor in a programmable oven, which was heated to 250 °C for 2 h with a heating rate of 3 °C min$^{-1}$, then allowed to cool naturally to room temperature. Finally, the clean colored CF fabric (7CC) was obtained by immersing it in the aqueous ammonia for 60 min (aqueous ammonia ($NH_3 \cdot H_2O$, 28%)), washed with distilled water, and dried in an oven at 80 °C. The preparation process of colored CF yarns was the same as that of the colored CF fabrics, while the mass of raw CF yarns was 5.8 g. Under the same reaction conditions, colored CF fabrics with different glucose concentrations of 4 g 70 mL$^{-1}$ (4CC), 5 g 70 mL$^{-1}$ (5CC), 6 g 70 mL$^{-1}$ (6CC), 8 g 70 mL$^{-1}$ (8CC), 9 g 70 mL$^{-1}$ (9CC), 10 g 70 mL$^{-1}$ (10CC), 11 g 70 mL$^{-1}$ (11CC), 12 g 70 mL$^{-1}$ (12CC), 13 g 70 mL$^{-1}$ (13CC), 14 g 70 mL$^{-1}$ (14CC), 15 g 70 mL$^{-1}$ (15CC), 16 g 70 mL$^{-1}$ (16CC), and 17 g 70 mL$^{-1}$ (17CC) were prepared. The residual solids and raffinate solution were separated by centrifugation. Meanwhile, after washing the residual solids using distilled water five times, the clean residue was obtained by drying at 80 °C.

### Characterization

The surface structures of colored CF fabric were characterized by the scanning electron micrograph (SEM, JSM-7800, and Zeiss Gemini 300) and Transmission electron microscope (TEM, JEM-2100F). The carbon content was conducted by the elemental analyzer (EA, Vario el cube). The crystalline structures of CSs in the residue after hydrothermal reaction were measured by an X-ray diffraction system (XRD, Empyrean). Molecular structures were obtained using a Fourier Transform

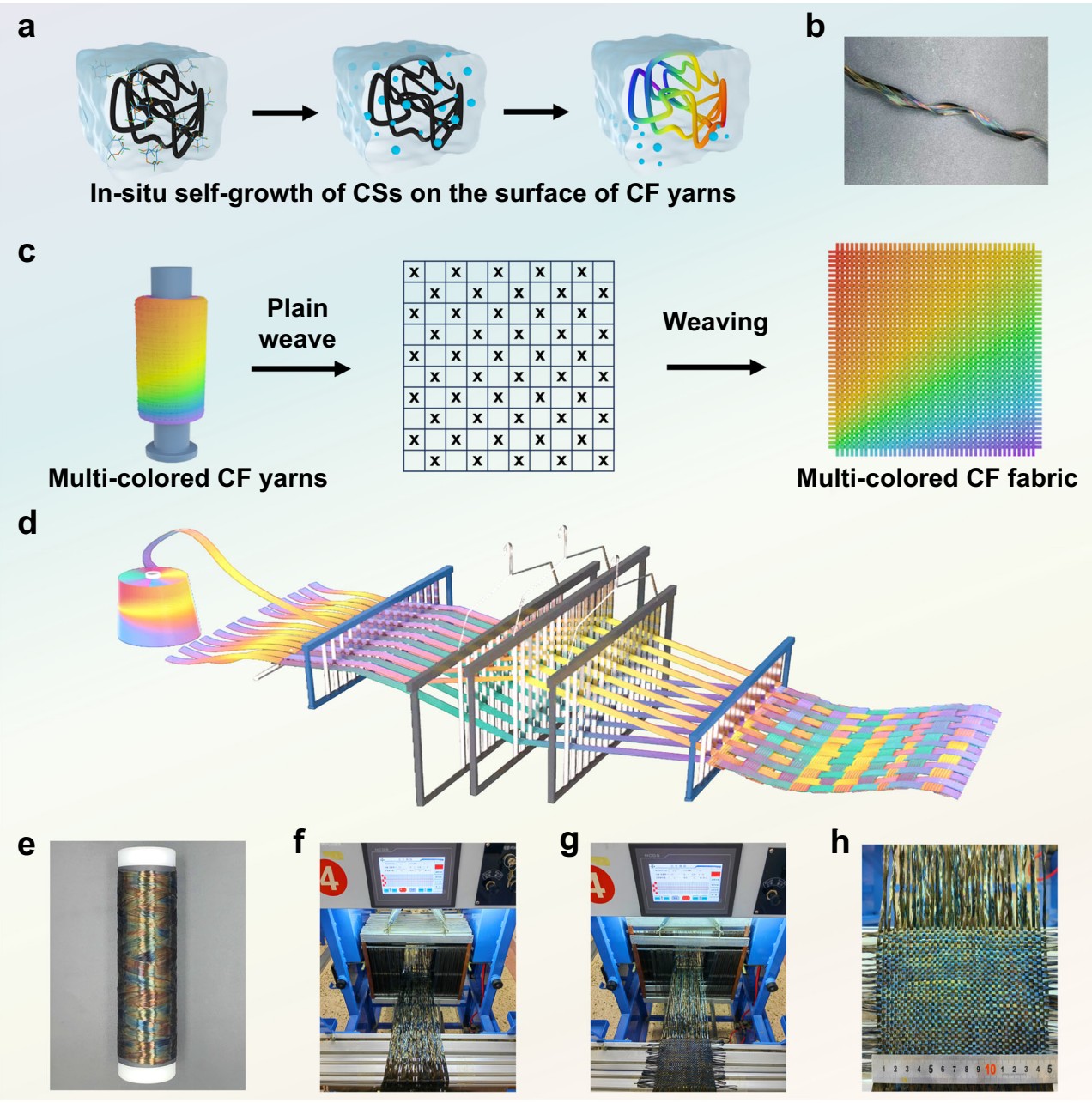

**Fig. 6 | Textile weaving of a plain multi-colored CF fabric on the weaving machine using multi-colored CF yarns. a** Multi-colored CF yarns prepared using the one-step HTC of glucose. **b** Photograph of a multi-colored CF yarn. **c** Schematic illustration of the production process of colored CF fabric from multi-colored CF yarn. **d** Weaving process of preparing colored CF fabric. **e** Photograph of a package of multi-colored CF yarn. **f** Photograph of the weaving machine. **g** Photograph for the weaving process of the multi-colored CF fabric. **h** A piece of 15 × 15 cm² plain weave fabric formed by industrial weaving machines.

Infrared Spectroscopy (FTIR, Nicolet iS50). Solid-state MAS NMR $^{13}$C spectra were acquired on a Solid Nuclear Magnetic Resonance (NMR, Agilent DD2-500MHz). Solution-state NMR $^{13}$C spectra were acquired on a Solution Nuclear Magnetic Resonance (NMR, Bruker Avance III HD 500 MHz). The *XPS* spectra were performed by X-ray photoelectron spectroscopy with Al Ka X-ray (*XPS*, ESCALAB 250Xi). The complex refractive index was performed by the Ellipsometer (Ellipsometer, J.A.Woollam M-2000). The simulated reflectance spectra were calculated with commercial finite-difference time-domain software (FDTD, Lumerical). The mass spectrum was run on the Matrix-assisted laser desorption ionization time-of-flight mass spectrometer (MALDI-TOF, Bruker auto flex III). The reflection spectra of colored CF fabric were measured by an Ultraviolet-visible spectrophotometer (UV-vis, UV3600). The chromaticity information was performed on the

Benchtop Spectrophotometer (Benchtop Spectrophotometer, Color i7). The optical photographs and three-dimensional optical pictures were collected with a Canon camera (Canon, EOS 760D) and a three-dimensional microscope (3D microscope, RH-2000). Angle-resolved reflective spectrum was employed to investigate the isotropic optical property (Angle-resolved Spectroscopy, ARMS). Rubbing fastness of structural colored fabrics was performed on a Rubbing color fastness meter (Crockmeter, YB571-II). The shaker was used in the acid pickling process (Shaker, HBC-24). The stability of structure color under intense light was characterized under a xenon lamp light source system (xenon lamp, CEL-S500). The light intensity was monitored by the Optical power meter (OPM, CEL-NP2000-2A). 2D Fourier transformation applied to SEM images was acquired from MATLAB software (MATLAB, 2018). The mechanical property was analyzed using a universal material testing

system (Instron, 5943). The electrical property was measured using the Graphical Series Source meter (Graphical Series Source meter, 2450).

## Data availability

All data generated in this study are provided in the article and its Supplementary Information. Source data are provided with this paper. Additional data are available from the corresponding author upon request. Source data are provided with this paper.

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

## Acknowledgements

This work was financially supported by the National Natural Science Foundation of China (U21A2095, W.X.; 52303064, L.X.); the Foundation of Science Research Program from the Hubei Provincial Department of Education (Q20221711, L.X.); the Natural Science Foundation of Hubei Province (2022CFC071, L.X.); the Key Research and Development Program of Hubei Province (2021BAA068, W.X.).

## Author contributions

S.J.Z., C.H.Z. and Z.F. fabricated samples. S.J.Z. and Q.M.Z. performed the characteristics examination. S.J.Z., Z.Z.X.Z. and J.Y.G. provided model analysis. S.J.Z., C.H.Z., Z.F., N.Z., X.F.W. and X.J.W. performed data analysis and discussed the results. L.J.X. and W.L.X. supervised the project. S.J.Z. and L.J.X. wrote the manuscript with input from all co-authors.

## Competing interests

The authors declare no competing interests.
