## [Peer Review File · Nature Communications]

REVIEWER COMMENTS

Reviewer #1 (Remarks to the Author):

The authors developed an in-situ self-growing strategy to prepare a coating of carbon spheres onto the carbon fibers, realizing the coloration of carbon fibers, which is sustainable and scalable compared to other methods. By changing the concentrations of glucose solution, the diameters of carbon nanospheres are controlled, achieving the multicolor structural coloration of carbon fibers. This in-situ growth of carbon spheres by one-step hydrothermal carbonation of glucose manner is very impressive and provides a sustainable way to achieve the large-scale preparation of colored carbon fibers with superior color stability. I strongly recommend the publication after the following minor revision.

1. There may be some unclear words in this article.

- Page 6, Figure 1, the "(d)" should be replaced by "(d) and (e)".
- Page 15, Line 7, the word "nation" in the sentence "These findings support the nation that CSs are a promising candidate for the structural color of CF fabrics, ..." may be not appropriate.
- Page 16, Line 1, the "Fig.5g" should be replaced by "Fig.4g".
- Page 18, Line 1, the word "lesion" should be replaced by "lesson".

2. Does the carbon spheres coating influence the intrinsic properties of carbon fibers, such as the superior electrical and mechanical properties?

3. Is there any clear quantitative correlation between the diameters of carbon spheres and the hues of structural coloration via scattering theory?

4. I am a little confused about the structural coloration of 17CC (green) with an average diameter of 242.4 nm. The reasons are as follows: Firstly, in Figure 1 (g), according to the trends of reflection spectral curves, the reflectance spectrum of 17CC seems to be formed by the redshift of 13CC. In that case, the diameter of 17CC should be larger than 13CC of 297.2 nm. According to Mie scattering theory, the higher-order scattering peaks are lower and wider, while the lower-order scattering peaks are higher and narrower. The higher-order scattering peak of 17CC gives it the green reflection peaks, which is consistent with its color hue but not consistent with its Mie scattering calculations. Secondly, the diameter of carbon spheres from 17CC is not consistent with the positive correlation that the size of the carbon spheres will increase with the increase of concentrations of glucose. And strangely, the average diameter of 17CC has dropped too much when the concentration of glucose is changed from 13CC to 17CC. In a word, I think that the real diameter of 17CC may be larger than 297.2 nm, and it may be in the range of 320~340 nm.

Reviewer #2 (Remarks to the Author):

In this manuscript, inspired by the disordered structure of peacock's back feathers, the authors develop a one-step hydrothermal process to realize the fabrication of colored carbon fiber (CF) fabric with the in situ randomly self-growing of carbon spheres (CSs), which solves the challenge of monochromatic color and inert property of traditional CF. The structural color is derived from the amorphous structure of CSs on the CF surface, which can be easily controlled by tuning the concentration of glucose solution as feeding materials. Moreover, the interfacial interaction between CSs and CF endows the structure with color stability, mechanical robustness, acid/alkali durability and light fastness, which provides a solution for coloration of high-performance fibers. The implementation of the proposed approach is interesting, however some statements in the manuscript are not well supported. Therefore, a revision is required before it could be published in Nature Communications.

1. As the authors mention, the structure of the colored CF fabric is mimicked the peacock's back feathers. However, the enlarged feather SEM image in Fig. 1d is blurred. In order to make a clear contrast, please provide a clearer SEM image of Fig. 1d to show its disordered morphological characteristic.
2. The mechanism of the distribution and size of CSs regulated by the glucose concentration is not demonstrated. The statement in Line 317 "The average grain diameters were measured to be 183.6 nm, 264.6 nm, 274.0 nm, 297.2 nm, and 242.4 nm for CSs on 4CC, 7CC, 10CC, 13CC, and 17CC CF fabrics", why does the average grain diameter of CSs decrease when a larger glucose concentration is used? Moreover, as shown in Table S1, the particle size distribution range of CSs is wide, will it affect the color purity of the final CF fabrics? It is suggested to characterize the full width at half maximum.
3. The statement in Line 175 "the color of the disordered structure was direction-independent", why do the images in Fig. 3 f-j being observed in 3D optical microscopy display different colors? It is confusing.
4. Please provide K/S spectrum after longer cycles, the result of 10 cycles may not be representative. In addition, to better and clearer illustrate the color fastness, acid pickling, and light fastness, it is suggested to supplement the corresponding optical images and K/S spectra of the fabricated CF fabrics before test in Fig. b, c, Fig. e, f, Fig. h, I and Fig. S5.
5. Some details need to be verified, revised and added.
 - (1) The statement in Line 337 "titled incident light in the microscope light path", it may be "tilted", rather than "titled".
 - (2) Abbreviations need to be spelled out for the first time, for example the K/S spectrum in Line 327.
 - (3) The statement in Line 349 "color fastness was evaluated by color fastness, acid pickling, and light fastness" is a little strange. The second "color fastness" may be "mechanical stability".
 - (4) The statement in Line 362 "Excellent acid and alkaline resistance were observed when combined with the cleaning process of the colored CF fabric using ammonium hydroxide" lacks corresponding data to support.
 - (5) The statement in Line 369 "After continuous intense light exposure for 60 min, the colored CF fabric showed a blue shift of the characteristic peak in the K/S spectrum", Please explain why the characteristic peak blue-shifts after light exposure.

Reviewer #3 (Remarks to the Author):

The authors have presented a study on the in-situ growth of carbon spheres (CSs) on the surface of carbon fibers (CF) using glucose as a feeding material. The randomly disordered structure of CSs, which is attributed to the Mie resonances of CSs of different sizes influenced by the glucose concentration, results in excellent optical coloration for CF fabrics and color stability. The interaction between CSs and CF promotes stable interfacial forces, contributing to the outstanding mechanical robustness, acid/alkali durability, and ultraviolet-visible color fastness of colorized CFs.

However, there are some critical concerns that need to be addressed. Firstly, there is a lack of clear explanation regarding the coloration mechanism of randomly arranged CSs on the CF surface and the reaction mechanism from glucose to CSs. Additionally, the data analysis presented is inconsistent and does not sufficiently support the conclusions drawn. These issues raise concerns about the overall quality and validity of the study. Therefore, the current manuscript is unacceptable for publication in Nature Communications.

Furthermore, there are several points that need improvement within the manuscript:

In the introduction, the authors review structural colors derived from periodic photonic crystals and amorphous photonic structures. They also mention the brilliant colors observed in the back feather of a peacock, which are supposedly originating from randomly disordered amorphous nanostructures. However, according to the literature [Wang et al., 2016; Yoshioka & Kinoshita, 2002], the yellow feather (Figure 1(a)) actually possesses a clear periodic photonic crystalline structure along the cross-

sectional direction under the surface of a barbule. Therefore, the analogy drawn to the structure of the peacock feather in this context is not well-founded. Moreover, the SEM photographs of the peacock feather in Figure 1(d) and Figure S6 show a smooth surface at different magnifications with 10 μ m and 100nm scale bars. Consequently, the randomly amorphous nanostructures claimed to exist on the peacock feather are not observed. To clarify the microscopic structure of the peacock feather, it would be beneficial to analyze the cross-sectional structure of a barbule. Additionally, it is important to note that the structure of CSs on the CF surface is unnecessary to mimic the structure of the peacock feather, as they share no similarities.

The hydrothermal reaction of glucose with carbon fibers/fabrics raises several unanswered questions. The assumed reaction mechanism involves the hydrolysis and dehydration processes of glucose, followed by the production of polymeric intermediates. These intermediates are hypothesized to nucleate and grow on the surface of CFs, eventually carbonizing and forming the firmly attached CSs on the CF surface. However, there is a lack of evidence to support this proposed mechanism. Furthermore, it is unclear what the chemical composition of the carbon spheres is, especially considering the difficulty in obtaining pure carbon spheres under the hydrothermal temperature of only 250°C. Additionally, if the carbon spheres were to nucleate and grow on the CF surface, one would expect their shape to be mostly hemispherical. However, the SEM images in Figure 2(a)-(d) show a large amount of spherical carbon spheres distributed and deposited on the CF surface, raising questions about the formation process.

The mechanism of structural coloration remains unclear but is of great importance, especially considering its attribution to the Mie scattering resonance of CSs in this study. Structural colors resulting from disordered structures possess isotropic optical properties, which can be characterized using angle-resolved reflective spectra. Furthermore, structural colors arising from Mie scattering resonance are highly dependent on the size and composition of the CSs. However, the manuscript does not present quantitative control over the size and composition of CSs. The distribution of CSs on the CF surface is theoretically expected to have a determinative effect on the resulting structural colors.

The physical implication and definition of the K/S value are unclear. While it is generally used for pigment dyeing and can be applied to characterize structural colors, its specific relevance to this study should be clarified. Additionally, it would be valuable to explain the changes in the structure of CSs after soaking and washing in an acetic acid solution under vibration for 120 minutes at a rate of 60 r/min for color fastness testing.

Figure 5 illustrates the weaving process of the segment-color yarn to create colored CF fabric. However, it is not explained how the colorful yarn is continuously produced through the hydrothermal methodology in a batch process. Furthermore, there are concerns regarding the potential damage to the photonic structure of CSs on the CF surface during the weaving process, which could eventually lead to deterioration of the fabric colors. These aspects need to be addressed and discussed in more detail.

The overall English expression in the manuscript requires significant improvement. There are numerous errors and inaccurate sentences that need to be corrected to enhance clarity and readability.

By addressing these concerns and making the suggested improvements, the manuscript could be significantly enhanced in terms of scientific rigor, clarity, and overall quality of presentation.

Dear Reviewers,

We sincerely thank Reviewers for taking time to review our manuscript 'NCOMM S-23-14057 A' entitled 'Color construction of multi-colored carbon fibers using glucose'. The professional and critical comments of Reviewers helped us to improve the quality of our manuscript. We have taken into account all the insightful comments of the Reviewers and revised the Manuscript and Supplementary Information accordingly. Our point-by-point reply are included in blue font after each reviewer comments that are available in black, and changes made to the manuscript are shown in red color. A clear revised version of the manuscript and a red-marked version with all changes clearly indicated in this submission.

We appreciate for Editor and Reviewers' warm work earnestly, and hope that these corrections will meet with approval.

Thanks again for your consideration of our manuscript!

Best regards,

Liangjun Xia

Reviewer #1

Comment summary: The authors developed an in-situ self-growing strategy to prepare a coating of carbon spheres (CSs) onto the carbon fibers, realizing the coloration of carbon fibers, which is sustainable and scalable compared to other methods. By changing the concentrations of glucose solution, the diameters of carbon nanospheres are controlled, achieving the multicolor structural coloration of carbon fibers. This in-situ growth of carbon spheres by one-step hydrothermal carbonation of glucose manner is very impressive and provides a sustainable way to achieve the large-scale preparation of colored carbon fibers with superior

color stability. I strongly recommend the publication after the following minor revision.

Reply: We would like to thank the expert Reviewer for your positive comments and insightful suggestions on our manuscript, which allowed us to achieve the current improved version. We have taken into account all the professional comments from you and revised the manuscript accordingly.

Comment 1. There may be some unclear words in this article.

Page 6, Figure 1, the “(d)” should be replaced by “(d) and (e)”.

Page 15, Line 7, the word “nation” in the sentence “These findings support the nation that CSs are a promising candidate for the structural color of CF fabrics, ...” may be not appropriate.

Page 16, Line 1, the “Fig.5g” should be replaced by “Fig.4g”.

On page 18, Line 1, the word “lesion” should be replaced by “lesson”.

Reply: Thank you very much for your careful reading! In the revised manuscript, according to the professional comments, we have amended the unclear words. Additionally, we have checked the manuscript to avoid unclear words.

In Figure 1, the “(d)” was corrected to “(d) and (e)”.

“These findings support the nation that CSs are a promising candidate for the structural color of CF fabrics, ...” has been corrected to “The process of color construction potentially promoted the color development of carbon materials.”

The “Fig.5g” has been corrected to “Fig.4g”.

The word “lesion” should be corrected to “lesson”.

The spelling mistake of “titled” has been corrected to “tilted”.

The word “endows” should be corrected to “endowing”.

“to design” has been corrected to “for designing”.

“has emerged as” has been corrected to “have emerged as”.

The word “choice” has been amended to “choices”.

The word “renewable” has been amended to “renewability”.

“compared to” has been corrected to “compared with”.

“color fastness” has been amended to “mechanical stability”.

The sentence “color fastness was evaluated by color fastness, acid pickling, and light fastness” has been corrected to “color fastness was evaluated by mechanical stability, acid pickling, and light fastness”.

“Inspired by the structure of brilliant peacock back feathers” has been amended to “inspired by the naturally grown protrusive nanostructures on the green central surface of peacock back feathers”.

“endows it with many outstanding properties of mechanical robustness” has been corrected to “endowing it with outstanding mechanical robustness”.

“inspired by the robust structural colors derived from back feather of peacock” has been amended to “inspired by the naturally grown protrusive nanostructures on the green central surface of peacock back feathers”.

“Inspired by the disordered structure of peacock’s back feathers” has been corrected to “inspired by the naturally grown protrusive nanostructures on the green central surface of peacock back feathers”.

We have carefully checked the revised manuscript and Supporting Information. The unclear and inaccurate words have been amended in the revised manuscript in red color.

We sincerely thank you again for your careful reading!

Comment 2. Does the carbon spheres coating influence the intrinsic properties of carbon fibers, such as the superior electrical and mechanical properties?

Reply: We would like to thank the Reviewer for the professional comments. To investigate the influence of carbon spheres (CSs) on the fibers, we have conducted the electrical and mechanical properties of carbon fibers (CFs) before and after the coloration process, which is shown below, and the discussions about the electrical and mechanical properties have been added in the file of **Supplementary Information**.

Appendix Fig.1 Current-voltage curves of fibers bundle driven from the raw carbon fiber (CF) fabric and colored CF fabric.

“The bundle of CFs is clamped to the electrodes with a test length of 5 cm, and the range of test voltage is set at 0.0–1.0 V. The operating process is shown in Supplementary Fig. 11a (Appendix Fig. 1a in this file). Herein, the bundle of CFs was derived from the raw and colored carbon fiber (CF) fabric. Supplementary Fig. 11b (Appendix Fig. 1b in this file) shows that the slope of current-voltage curves reduced for the colored carbon fiber, which demonstrates that the electrical conductivity of raw CFs is better than that of colored CFs. This may be mainly attributed to the increase in the electrical contact resistance of the colored fiber surface. Additionally, the stable surface contact between CSs and CFs may also contribute to this phenomenon. As demonstrated in Supplementary Fig. 12a (Appendix Fig. 2a in this file), the mechanical performance of the CFs bundle, which was driven from the raw or colored CF fabric, was tested by Instron. The CSs coatings exhibited an enhancement influence on the mechanical performance of colored CFs. As shown in Supplementary Fig. 12b (Appendix Fig. 2b in this file), compared with pristine

CFs, the tensile strength of colored CFs was enhanced.”

Appendix Fig.2 Mechanical properties of carbon fibers (CFs) bundle driven from the raw CF fabric and colored CF fabric.

We have re-written this part according to the Reviewer’s professional suggestions, which have been added in the section of “**Results**”.

“Furthermore, the electrical and mechanical properties of different colored CF fabric were investigated. Compared with the raw CFs bundle, the surface contact resistance of colored CFs enhanced (Supplementary Fig. 11), owing to the stable contact between CSs and the surface of CFs. As demonstrated in Supplementary Fig. 12, the mechanical performance of the colored CF bundle was enhanced with the obvious increment of tensile strength.”

Appendix Fig.1 and Appendix Fig.2 have been revised as Supplementary Fig. 11 and Supplementary Fig. 12, which have been added in the section of “**Supplementary Information**”.

Thank you for your insightful comments!

Comment 3. Is there any clear quantitative correlation between the diameters of carbon spheres and the hues of structural coloration via scattering theory?

Reply: We sincerely think this is an excellent suggestion. We have conducted experiments for investigating the quantitative correlation between the diameters

of CSs and the hues of structural coloration, including the exact location where the change can be found in the revised manuscript. Indeed, the structural colors of CF fabrics are also related to the diameters of CSs, which was clarified by the curve that appeared in Appendix Fig.3 obtained from experimental results by Gaussian fitting for the distribution of diameter. We measured the diameters of CSs on the colored CF fabrics by Image J software, which were derived from the various glucose concentrations of 4CC, 7CC, 10CC, 13CC, and 17CC at the same hydrothermal reaction conditions. Then, the reflection spectra of 4CC, 7CC, 10CC, 13CC, and 17CC were employed to investigate the characteristic wavelengths, which were shown in Appendix Fig.4.

The explanations for the correlation between the diameters of particles and the hues of structural coloration have been proposed. Tuning the scattering properties of materials in controlling size can achieve color control. Magkiriadou et al.¹ used a quantitative scattering model to propose that color appearance is dominated by backscattering resonances of individual particles. Schertel et al.² obtain blue, green, red, and purple colors repeatably by varying the polystyrene colloidal scatterer size. Moreover, this research proved that the particle size determined the color hue, and with the variation of particle size, the color was changed, in which blue is only possible for the smallest particle sizes. Furthermore, Mao et al.³ proposed a system-disordered nanostructure with a material-independent optical response. For neglecting variation, the indiscernible structural color was generated by the substitution of different materials at the same particle size. And Chu et al.⁴ proved that the hue could be tuned by controlling the size of the spheres in the amorphous photonic structures. Additionally, as investigated by Cho et al.⁵, the green color corresponded to the largest particle size developing from the peak of the reflectance spectra, compared to the red color from the dip of the reflectance spectra. Therefore, in this research, the characteristic wavelength was recorded at 400, 430, 470, 540, and 570nm for 4CC (blue), 7CC (yellow), 10CC (orange-

red), 13CC (purple), and 17CC (green) CF fabrics, according to the peak(4CC and 17CC), and dip (7CC, 10CC, and 13CC) of reflection spectra (Appendix Fig. 4a), which was consistent with the wavelength corresponding to the maximum K/S value (Appendix Fig. 4b) (K/S value is the ratio of the absorption coefficient to the scattering coefficient for an opaque object, and a K/S value indicates the depth of color). The investigation of the relationship between average particle size and characteristic wavelength was shown in Appendix Fig. 3f.

Appendix Fig. 3 The diameter distribution for CSs nanoparticles exhibiting different colors on CF fabric, which are, from **a** blue, **b** yellow, **c** red, **d** purple, and **e** green. **f** measured characteristic wavelengths as a function of the average CSs diameters.

Appendix Fig. 4 a Reflection spectra, and **b** K/S curves of 4CC, 7CC, 10CC, 13CC, and 17CC.

According to the professional suggestion from the Reviewer, we have added the relationship between average particle size and characteristic wavelength in

the section of discussion in the revised manuscript, which was shown as follows. “Moreover, the characteristic wavelengths of colored CF fabrics in the reflectance spectra were recorded at 400, 430, 470, 540, and 570 nm, respectively, demonstrating for blue (400 nm), yellow (430 nm), orange-red (470 nm), purple (540 nm), and green (570 nm), according to the peaks (4CC and 17CC) and dips (7CC, 10CC, and 13CC) in Fig. 1g (Appendix Fig. 4a in this file). As shown in Supplementary Fig. 3 (Appendix Fig. 4b in this file), the K/S value at the characteristic wavelength was used to assess the color stability, and the characteristic peaks in K/S spectra correspond to the maximum K/S values at 700, 430, 470, 540, and 700 nm, matching with the dips of reflectance spectra. The diameter distribution and representative average diameter of CSs were estimated using the Gaussian function based on the diameter measurement of CSs on the colored CFs (Supplementary Fig. 4) (Appendix Figs. 3a–e in this file). The CSs with average diameters of 212.0, 258.7, 282.9, 308.6, and 350.0 nm on the CFs contributed to the blue, yellow, orange-red, purple, and green structural colors, respectively. The relationship curve between the characteristic wavelengths and CSs diameters for various colors is displayed in Fig. 1h (Appendix Fig. 3f in this file), further demonstrating that the average diameter of CSs on the CFs can tune the diversity of colors by controlling the concentration of the glucose solution.”

The diameter distribution for CSs nanoparticles of various colored CF fabrics has been nominated as Supplementary Fig. 4 in the section of “**Supplementary Information**”.

The reflection spectra and K/S curves have been nominated as Fig. 1g and Supplementary Fig. 3 in the section “**Results**” and “**Supplementary Information**”, respectively.

The relation curve between the characteristic wavelengths and CSs diameters for the various colors has been nominated as Fig. 1h in the section of “**Results**”.

Thank you very much again for the insightful comments from the Reviewer!

Comment 4. I am a little confused about the structural coloration of 17CC (green) with an average diameter of 242.4 nm. The reasons are as follows: Firstly, in Figure 1 (g), according to the trends of reflection spectral curves, the reflectance spectrum of 17CC seems to be formed by the redshift of 13CC. In that case, the diameter of 17CC should be larger than 13CC of 297.2 nm. According to Mie scattering theory, the higher-order scattering peaks are lower and wider, while the lower-order scattering peaks are higher and narrower. The higher-order scattering peak of 17CC gives it the green reflection peaks, which is consistent with its color hue but not consistent with its Mie scattering calculations. Secondly, the diameter of carbon spheres from 17CC is not consistent with the positive correlation that the size of the carbon spheres will increase with the increase of concentrations of glucose. And strangely, the average diameter of 17CC has dropped too much when the concentration of glucose is changed from 13CC to 17CC. In a word, I think that the real diameter of 17CC may be larger than 297.2 nm, and it may be in the range of 320~340 nm.

Reply: Thanks for this critical comment on our work. We were sorry for our careless mistakes. As the Reviewer pointed out, based on the trends of reflection spectral curves, the diameter of 17CC should be larger than 13CC of 297.2 nm.

Indeed, according to Mie scattering theory, the higher-order scattering peaks are lower and wider, while the lower-order scattering peaks are higher and narrower. Cho et al.⁵ investigated the correlation between color and particle diameter. With the increment of particle diameter to 360 nm, the transformation of structural color was demonstrated from red to green. The same pattern was proved by Mao et al.³, which demonstrated that the green color would be

exhibited with the nanoparticle diameter increasing at 330 nm. The estimation of average CSs diameter was limited by the range of measurement. Therefore, for the comment raised by the Reviewer, in the revised manuscript, the CS diameter was measured by a large number of samples at different regions to improve the stable distribution, and a representative average diameter was estimated by Gaussian Function. As shown in Supplementary Fig. 4 (Appendix Fig. 5a in this file), the measured average CSs size of 17CC colored CF fabrics was 350.0 nm. The relation curve between the characteristic wavelength and CSs size for the various colors was displayed in Fig. 1 h (Appendix Fig. 5b in this file), in which the linear relationship between the measured characteristic wavelength and CSs has been demonstrated, and the average CSs diameters on the CF fabrics were controlled by tuning the concentration of glucose solution as feeding materials.

Appendix Fig. 5 **a** Diameter distribution for CSs nanoparticles of 17CC (green). **b** Measured characteristic wavelengths as a function of the average CSs diameters.

Thank you very much again for your positive comments and valuable suggestions to improve the quality of our manuscript. We have made extensive modifications to our manuscript and supplemented extra data to make our results convincing.

Thank you very much!

Reviewer #2:

Comment summary: In this manuscript, inspired by the disordered structure of peacock's back feathers, the authors develop a one-step hydrothermal process to realize the fabrication of colored carbon fiber (CF) fabric with the in situ randomly self-growing of carbon spheres (CSs), which solves the challenge of monochromatic color and inert property of traditional CF. The structural color is derived from the amorphous structure of CSs on the CF surface, which can be easily controlled by tuning the concentration of glucose solution as feeding material. Moreover, the interfacial interaction between CSs and CF endows the structure with color stability, mechanical robustness, acid/alkali durability, and light fastness, which provides a solution for the coloration of high-performance fibers. The implementation of the proposed approach is interesting, however, some statements in the manuscript are not well supported. Therefore, a revision is required before it could be published in Nature Communications.

Reply: We sincerely thank the Reviewer for the careful reading and positive comments on our manuscript. Meanwhile, we feel great thanks for your professional review work and insightful suggestions to improve our manuscript. We agree with all your professional comments, and we have corrected the manuscript point by point accordingly.

Comment 1. As the authors mention, the structure of the colored CF fabric is mimicked the peacock's back feathers. However, the enlarged feather SEM image in Fig. 1d is blurred. To make a clear contrast, please provide a clearer SEM image of Fig. 1d to show its disordered morphological characteristic.

Reply: We sincerely appreciate the valuable comments. As the Reviewer

pointed out, we agree with the constructive suggestions that the enlarged SEM images of the feather in Fig. 1 should be provided to show the morphological characteristic, which was important for explaining the mimic. We have conducted extensive surface analyses on the back feathers of the peacock by using SEM. The high-resolution SEM images of the feather have been replaced in Fig. 1, and we have also added clearer SEM images for the surface structure of the feather in Supplementary Fig. 1 (Appendix Fig. 6 in this file), which is shown below.

Appendix Fig. 6 **a** Photograph of the peacock back feather. **b** The angle-resolved reflectance spectroscopy of the central area (green) for the peacock back feather. **c-h** SEM images of the barbule surface at the central area of the peacock back feather.

As shown in Appendix Figs. 6c-h, the obvious nanoparticles were observed on the surface of the Peacock back feather with the resolution of the SEM images increased. In the revised manuscript, we have amended the corresponding sentence, which is shown below.

A striking structure has been created by living organisms on the center area of

the peacock back feather, exhibiting the unique characteristics of angle-independent vivid green structural color (Appendix Figs. 6a, b). Additionally, some spherical nanoparticles could also be observed on the surface with the resolution of the SEM images increased (Appendix Figs. 6f-h). Inspired by this nanostructure on the surface formed during the natural growth of the feather, a strategy of the randomly dynamic in situ growth of bio-based CSs was proposed for the color construction of structural color on CF fabrics.

This part has been amended in the section of “**Results**”, which is shown below.

“A striking structural color has been created by living organisms on the central area of the peacock back feather, exhibiting the unique characteristic of angle-independent property (Fig. 1a) (Appendix Fig. 6a, b in this file). Meanwhile, as shown in Fig. 1b and Supplementary Fig. 1 (Appendix Figs. 6c-h in this file), on the barbule surface for the central area of the feather, the protrusive nanostructures are formed during the natural growth of the feather. Inspired by this protrusive nanostructure, we proposed a strategy for the random in-situ growth of CSs on the CF fabric only using glucose as the feeding material (Fig. 1c, d).”

The SEM images with higher resolutions for the surface structure of the peacock back feathers have been added in the section of “**Supplementary information**”.

Thank you very much again for your valuable suggestions to improve the quality of our manuscript.

Comment 2. The mechanism of the distribution and size of CSs regulated by the glucose concentration is not demonstrated. The statement in Line 317 “The average grain diameters were measured to be 183.6 nm, 264.6 nm, 274.0 nm, 297.2 nm, and 242.4 nm for CSs on 4CC, 7CC, 10CC, 13CC,

and 17CC CF fabrics”, why does the average grain diameter of CSs decrease when a larger glucose concentration is used? Moreover, as shown in Table S1, the particle size distribution range of CSs is wide, will it affect the color purity of the final CF fabrics? It is suggested to characterize the full width at half maximum.

Reply: Thank you very much for raising this question. Firstly, we sincerely apologize that we did a measurement error for the average diameter of CSs on the 17CC CF fabrics. Indeed, owing to the limitation of the measured quantity and range of the CSs on the CF fabrics, the reliable estimation of average diameter was limited. Therefore, as suggested by the insightful comment from the Reviewer, in the revised manuscript, the average diameter was assayed by a large number of samples at the different regions for the same color to improve the stable distribution. Then, the representative average particle diameter was estimated by Gaussian Function, which was characterized by the full width at half maximum displayed in Appendix Fig. 3. The measured average CSs diameters of 4CC, 7CC, 10CC, 13CC, and 17CC were 212.0, 258.7, 282.4, 308.6, and 350.0 nm, respectively.

During the construction of structural color on CF fabrics, the concentration of glucose was investigated as the only variate to regulate the size of CSs. Furthermore, the correlation between characteristic wavelengths of various color and average CSs diameter has been constructed, which was shown in Appendix Fig. 3f. The corresponding part has been amended in the revised manuscript.

Appendix Fig. 3 The diameter distribution for CSs exhibiting different colors on CF fabric, which are, from **a** blue, **b** yellow, **c** orange-red, **d** purple, and **e** green. **f** measured characteristic wavelengths as a function of the average CSs diameters.

Therefore, in the revised manuscript, we rewrote the corresponding content:

“The diameter distribution and representative average diameter of CSs were estimated using the Gaussian function based on the diameter measurement of CSs on the colored CFs (Supplementary Fig. 4) (Appendix Figs. 3a–e in this file). The CSs with average diameters of 212.0, 258.7, 282.9, 308.6, and 350.0 nm on the CFs contributed to the blue, yellow, orange-red, purple, and green structural colors, respectively. The relationship curve between the characteristic wavelengths and CSs diameters for various colors is displayed in Fig. 1h (Appendix Fig. 3f in this file), further demonstrating that the average diameter of CSs on the CFs can tune the diversity of colors by controlling the concentration of the glucose solution.”

This part has been amended in the section “**Results**”.

Appendix Fig. 7 CIE diagram showing the color shift of 4CC, 7CC, 10CC, 13CC, and 17CC for the diameter change with respective x–y coordinates of (0.2716, 0.3019), (0.3909, 0.3814), (0.3955, 0.3600), (0.3276, 0.3165), and (0.3436, 0.3732), respectively.

Moreover, the color purity was visualized using an International Commission on Illumination (CIE) chromaticity diagram⁶. According to the critical comments of the Reviewer, the color purity of the colored CF fabrics was exhibited by CIE coordinates, and the CIE chromaticity diagram was emphasized in Fig. 1f (Appendix Fig. 7 in this file) of the revised manuscript. In the CIE chromaticity diagram, the CIE coordinates, close to the boundary of the chromaticity diagram, respond to ultrahigh color purity⁷. In fact, from a physical point of view, Kinoshita et al.⁸ shed light on that non-iridescent structural colors could be produced, which originated from irregularities of the structure. Indeed, the structural colored CF fabric shown by the CSs were composite colors formed by the combination of primary colors. As investigated by Chang et al.⁹, the vivid colors of patches for *Pachyrhynchus sarcitis* weevils derived from the reflection of differently oriented 3D photonic crystals within their scales, and the tuning of the distribution of the domain orientations could achieve different colorations. As mentioned in the research of Zhao et al.¹⁰, the structural colors shown by the SiO₂-coated carbon nanotube fibers (CNTFs) were composite colors formed by the combination of two or more primary colors. For structural color, the color coordinates of the CIE chromaticity diagram exhibited a similar region

for the color purity ^{6, 10}.

This part has been explained in the section “**Results**”.

We are sincerely thanking you for your professional comments!

Comment 3. The statement in Line 175 “the color of the disordered structure was direction-independent”, why do the images in Fig. 3 f-j being observed in 3D optical microscopy display different colors? It is confusing.

Reply: We sincerely appreciate the constructive comments. We also thank the Reviewer that you can give us a valuable chance to explain this phenomenon. There are several possible reasons resulting in the above phenomenon mentioned by the Reviewer.

Firstly, according to the previous investigations, for instance, the SiO₂-coated CNTFs exhibited brilliant colors, in which the photos exhibited different colors, which is mainly due to the structural colors consisting of composite colors formed by the combination of two or more primary colors ¹⁰. The optical microscopy of structurally colored chitin nanocrystal films demonstrated the composite color, which was fabricated by exploiting the self-assembly ⁶. For the creatures of nature, microscope pictures of colored patches for *Pachyrhynchus sarcitis* weevils showed different colored areas with colors in the entire visible spectral range ⁹. Furthermore, we would like to explain that, for high-resolution color, the high luminance LED light was coupled into a 3D optical microscope and focused onto samples by an objective lens with zooming in 1050 times.

To further investigate the color consistency, the different positions of colored CF fabrics have been measured, which is shown in Appendix Fig. 8. Also, to verify the direction-independent of colored CF fabrics, the angle-resolved reflectance spectroscopy has been investigated, which is shown in Appendix Fig. 8c,

displaying the obvious angle-independent structural color of the as-prepared fabrics.

Appendix Fig. 8 **a** K/S spectra at three different positions on 4CC, 7CC, 10CC, 13CC, and 17CC, respectively. **b** Maximum K/S values at characteristic wavelength according to the K/S spectra at three different positions and the average of K/S. **c** Angle-resolved reflectance spectroscopy of the colored CF fabric (7CC as the representative). **d-h** K/S spectra of the fabricated CF fabrics and **i** K/S value at characteristic wavelengths at three different positions after the mechanical rubbing, soaking and washing of acetic acid solution, and accelerated light-aging test for 4CC, 7CC, 10CC, 13CC, and 17CC (T-, A-, and L- were the colored CF fabric after mechanical rubbing test, acid pickling, and accelerated light-aging test, successively).

According to the professional comments from the Reviewer, we have added this part in the revised manuscript, which is shown below.

“As shown in Fig. 3k (Appendix Fig. 8a in this file), the colored CF fabric displays a relatively uniform distribution of K/S curves at different positions, essentially illustrating a homogeneous color of CF fabric. As expected, Fig. 3l (Appendix Fig. 8b in this file) shows the stable K/S values at characteristic wavelengths, representing the color depth, demonstrating its consistency in color appearance. To further investigate the directional independence of colored CF fabrics, the

angle-resolved reflection spectra were investigated. As shown in Fig. 3m (Appendix Fig. 8c in this file), the colored CFs fabrics exhibit the same reflection peaks when the fixed angle of the reflection source was adjusted from 30° to 150°, which illustrates an obvious angle-independent structural color. Therefore, the developed color construction process may potentially promote the stable color of CFs.”

“As shown in Figs. 4d–h (Appendix Figs. 8d–h in this file), the stable colors of CF fabrics were demonstrated by the no-obvious change of K/S spectra and K/S values at the characteristic wavelengths. To further investigate the color consistency, the average K/S values were calculated from the K/S spectra at three different positions of the colored CF fabric after mechanical rubbing, soaking and washing with the acetic acid solution, and accelerated light-aging. The appreciable stability of the K/S values (Fig. 4i and Supplementary Table 3) (Appendix Fig. 8i in this file) at characteristic wavelengths and the color presentation of the photographs (Supplementary Fig. 10) for 4CC, 7CC, 10CC, 13CC, and 17CC demonstrated excellent resistance to mechanical and chemical treatments of the as-prepared CF fabrics.”

The angle-resolved reflectance spectroscopy of colored CF fabric has been added in **Fig. 3** of the revised manuscript.

This discussion of direction-independent of colored CF fabric and the investigation of color consistency has been added in the section of “**Results**”.

Thank you very much!

Comment 4. Please provide the K/S spectrum after longer cycles, the result of 10 cycles may not be representative.

Reply: We sincerely thank the Reviewer for careful reading. According to the insightful comments from the Reviewer, we measured K/S values of the colored CF fabrics after various cycles of 0, 10, and 30 times, which demonstrated the

binding stability of CSs and the strength of color after different times of mechanical rubbing. Moreover, to accurately express the stability of the hue for as-prepared colored CF fabrics, the reflection spectra of these samples were investigated. According to ISO 105-X12, to indicate the mechanical stability of structural colors, the rubbing fastness testing was performed using a crock meter with 10 rubbing cycles under a 50 kPa load pressure ¹¹. Moreover, to display the excellent mechanical stability of colored CF fabrics, the reflection spectra and K/S values of the fabrics after 30 rubbing cycles were also measured, which is shown below.

Appendix Fig. 9 Color stability after mechanical rubbing cycles of 0, 10, and 30 times. The measured reflectance spectra of colored CF fabrics for **a** 4CC, **b** 7CC, **c** 10CC, **d** 13CC, and **e** 17CC. **f** K/S values of these samples (Tx; T: mechanical rubbing test; x: the cycle times).

Appendix Fig. 10 3D optical microscope images after friction resistance test of cycling 0 (T0), 10 (T10), and 30 (T30) times for 4CC, 7CC, 10CC, 13CC, and 17CC.

Appendix Fig. 9 and Appendix Fig. 10 are nominated as Supplementary Fig. 14 and Supplementary Fig. 15, which have been added in the section of “**Supplementary Information**”.

We have revised this part in the section of “**Supplementary Information**”, which is shown below.

“To verify the stable hue of the various colors, the structurally colored CF fabrics were performed by the color fastness friction meter with a load pressure of 50 kPa for 0, 10, and 30 cycles, respectively, and the reflection spectra before and after the mechanical rubbing test is shown in Supplementary Fig. 14 (Appendix Fig. 9 in this file). Also, the K/S values were measured from the characteristic peaks at three different positions on the colored CF fabrics after the rubbing cycles, which demonstrated that the various colored CF fabrics exhibited satisfactory fastness to the mechanical rubbing process. Furthermore, as shown in Supplementary Fig. 15 (Appendix Fig. 10 in this file), the 3D optical microscope images after the friction resistance test of cycling 0 (T0), 10(T10), and 30 (T30) times for 4CC, 7CC, 10CC, 13CC, and 17CC were measured. Compared with the original samples of cycling 0 (T0), the structure and color remained stable after the mechanical rubbing test.”

This part has been amended in the section of “**Results**”.

“To further demonstrate the color stability of the samples, we conducted a mechanical rubbing test at a load pressure of 50 kPa for 30 cycles, the reflectance spectra (Supplementary Figs. 14a–e), K/S values (Supplementary Fig. 14f and Supplementary Table 4), and 3D optical microscope images (Supplementary Fig. 15) demonstrated that the colored CF fabrics exhibited satisfactory fastness to the mechanical rubbing of 30 cycles.”

In addition, to better and clearer illustrate the color fastness, acid pickling, and light fastness, it is suggested to supplement the corresponding optical images and K/S spectra of the fabricated CF fabrics before testing

in Fig. b, c, Fig. e, f, Fig. h, I and Fig. S5.

Reply: We sincerely appreciate the valuable comments from the Reviewer. According to the insightful suggestions from the Reviewer, we have supplemented the corresponding optical images and K/S spectra of the fabricated CF fabrics after the color fastness, acid pickling, and light fastness, respectively, which is shown in Supplementary Fig. 10. (Appendix Fig. 11 in this file) Meanwhile, according to the Reviewer's suggestion, to better and clearer illustrate the color fastness, acid pickling, and light fastness, the K/S spectra of the colored CF fabrics were also summarized. In the revised manuscript, Fig. 4 (Appendix Fig. 12 in this file) has been amended to better and clearer illustrate the mechanical rubbing, acid pickling, and accelerated light-aging test for 4CC, 7CC, 10CC, 13CC, and 17CC, which demonstrated the stable structural color of the colored CF fabric.

Appendix Fig. 11 Photographs of the fabricated CF fabrics after the mechanical rubbing, acid pickling, and accelerated light-aging test for 4CC, 7CC, 10CC, 13CC, and 17CC.

Appendix Fig. 11 has been nominated as Supplementary Fig. 10 and added to the section of “**Supplementary Information**”.

The K/S spectra of the fabricated CF fabrics and K/S value at characteristic wavelengths after the mechanical rubbing, acid pickling, and accelerated light-aging test for 4CC, 7CC, 10CC, 13CC, and 17CC have been corrected in Fig. 4 (Appendix Fig. 12 in this file), which is shown below.

Appendix Fig. 12 **a** Schematic illustration of the mechanical rubbing test performed by color fastness friction meter with a load pressure of 50 kPa for 10 cycles. **b** Schematic illustration of soaking and washing of acetic acid solution test after mechanical rubbing test at a vibration of 60 times per minute for 120 min. **c** Schematic illustration of the accelerated light-aging test under the strong light using a xenon lamp light source system after mechanical rubbing test and soaking and washing of acetic acid solution (environment temperature, humidity, and light irradiance were set as 38 °C, 47 % RH, and $1.271 \times 10^4 \text{ W m}^{-2}$, respectively). **d-h** K/S spectra of the fabricated CF fabrics and **i** K/S value at characteristic wavelengths at three different positions after the mechanical rubbing, soaking and washing of acetic acid solution, and accelerated light-aging test for 4CC, 7CC, 10CC, 13CC, and 17CC (T-, A-, and L- were the colored CF fabric after mechanical rubbing test, acid pickling, and accelerated light-aging test, successively.).

Thank you very much again for your valuable suggestions to improve the quality of our manuscript!

Comment 5. Some details need to be verified, revised, and added.

(1) The statement in Line 337 “titled incident light in the microscope light path”, may be “tilted”, rather than “titled”.

Reply: Thank you very much for your careful reading and critical comments. We apologized for the spelling mistake of “titled”. In the revised manuscript, the “titled” has been corrected to “tilted”.

(2) Abbreviations need to be spelled out for the first time, for example, the K/S spectrum in Line 327.

Reply: According to the professional suggestion of the Reviewer, in the revised manuscript, abbreviations were spelled out for the first time at the corresponding position.

Additionally, the revised manuscript was proofread to correct the remaining mistakes, discourse errors, and writing inconsistencies.

(3) The statement in Line 349 “color fastness was evaluated by color fastness, acid pickling, and light fastness” is a little strange. The second “colorfastness” may be “mechanical stability”.

Reply: Thanks very much for the constructive opinion, in the revised manuscript, the second color fastness of the sentence has been corrected by mechanical stability, which is shown below.

“Furthermore, color fastness was evaluated by mechanical stability, acid pickling, and light fastness.”

(4) The statement in Line 362 “Excellent acid and alkaline resistance were observed when combined with the cleaning process of the colored CF fabric using ammonium hydroxide” lacks corresponding data to support it.

Reply: We appreciate the professional comments of the Reviewer, which help us to improve the quality of this manuscript.

K/S and reflection spectra were employed to investigate the excellent acid and alkaline resistance combined with the cleaning process and washing of acetic acid solution test for the colored CF fabric. Compared with the raw colored CF fabric, the reflection spectra and K/S spectra after cleaning with Ammonium

hydroxide, and acid pickling by Acetic acid for 7CC, indicate no apparent changes of color hue and strength, which is shown in Appendix Fig. 13.

Appendix Fig. 13 a Reflectance spectra and b-d the K/S spectra of raw colored CF fabric, cleaning with Ammonium hydroxide, and soaking and washing by Acetic acid for 7CC, respectively, as the representative sample.

Moreover, the definition of the value of K/S has been added in the revised manuscript.

“K/S value is a function ($K/S=(1-R)^2 / 2R$) developed by Kubelka and Munk, who theorized in 1931 that the ratio of the coefficient of light absorption (K) to the coefficient of light scattering (S) is corresponding to the fractional reflectance of the light (R) of the opaque substrate at a given wavelength^{12, 13}. Here, the K/S values of the colored CF fabrics could be directly measured by spectrophotometer at λ_{max} , which was presented with $K/S=(1-R_{\lambda_{max}})^2 / 2R_{\lambda_{max}}$. Therefore, the wavelength corresponding to the maximum K/S value (Supplementary Fig. 3) (Appendix Fig. 23 in this file), matching with the characteristic wavelengths of reflectance spectra, represented the color

strength used to assess the stability of color strength after different treatments.”

This part has been added in the section of “**Supplementary Information**”.

According to the suggestion of the Reviewer, considering that excellent acid resistance has been estimated in the process of soaking and washing with acetic acid (acetic acid (CH_3COOH , 50 wt%)) water solution under vibration of 60 times per minute for 120 min. We have added the evaluation of color stability with the clearing process using ammonium hydroxide in the section of “**The preparation of the color carbon fiber fabric**” and “**Supplementary Information**”, which is shown below.

Appendix Fig. 14 a Reflectance spectra of raw colored CF fabric, cleaning with Ammonium hydroxide; b and c K/S spectra of raw colored CF fabric, cleaning with Ammonium hydroxide for 7CC as the representative sample.

To further verify the color stability, the as-prepared samples were experienced in the washing process using ammonium hydroxide solution. The reflection spectra and K/S spectra were measured to investigate the hue and color strength of 7CC colored CF fabric as the representative sample. As exhibited in Appendix Fig. 14, compared with the 7CC colored CF fabric before the washing process, the reflection spectra and K/S values corresponding to the characteristic wavelength in K/S spectra after washing with ammonium hydroxide indicated no apparent changes, which elucidated the stable color hue and color strength.

(5) The statement in Line 369 “After continuous intense light exposure for 60 min, the colored CF fabric showed a blue shift of the characteristic

peak in the K/S spectrum”, Please explain why the characteristic peak blue-shifts after light exposure.

Reply: we are sorry for the incomplete expression of “the colored CF fabric showed a blue shift of the characteristic peak in the K/S spectrum, after continuous intense light exposure for 60 min”. We have corrected the presentation of this sentence in the revised manuscript.

The characteristic peak of the maximum value for K/S in the spectrum remains stable, but the K/S value corresponding to the short wavelength adjacent to the characteristic peak was enhanced, showing in Appendix Fig. 15. The real-time temperature of the colored CF fabric with continuous intense light exposure for 60 min was recorded, which rapidly increased to ~110 °C (Appendix Fig. 16). The particle sizes were reduced slightly upon enhancement of temperature, which did not play a vital role in the size¹⁴. Therefore, after continuous intense light exposure for 60 min, the color remained stable.

Appendix Fig. 15 K/S spectra of the colored CF fabric after continuous intense light exposure for 60 min.

Appendix Fig. 16 The real-time temperature of the colored CF fabric with continuous intense light exposure for 60 min.

To further investigate the color stability of the colored CF fabric after continuous intense light exposure for 60 min, the K/S spectra of the three positions for each sample were measured. The K/S spectra and the maximum values of K/S at characteristic peaks were displayed in Appendix Fig 17, which demonstrated the stable color.

Appendix Fig 17 a-e K/S spectra of the fabricated CF fabrics and **f** K/S value at characteristic wavelengths at three different positions after the mechanical rubbing, soaking and washing of acetic acid solution, and accelerated light-aging test for 4CC, 7CC, 10CC, 13CC, and 17CC (T-, A-, and L- were the colored CF fabric after mechanical rubbing test, acid pickling, and accelerated light-aging test, successively.).

The K/S spectra of the fabricated CF fabrics and K/S value at characteristic wavelengths after the mechanical rubbing, acid pickling, and accelerated light-aging test for 4CC, 7CC, 10CC, 13CC, and 17CC have been corrected in **Figs. 4d-i**.

Thanks again for carefully reading our manuscript and providing these professional and constructive comments, which are very helpful to improve the quality of our manuscript. We have carefully revised the manuscript according to your comments and corrected the remaining mistakes and writing inconsistencies.

Reviewer #3 (Remarks to the Author):

The authors have presented a study on the in-situ growth of carbon spheres (CSs) on the surface of carbon fibers (CF) using glucose as a feeding material. The randomly disordered structure of CSs, which is attributed to the Mie resonances of CSs of different sizes influenced by the glucose concentration, results in excellent optical coloration for CF fabrics and color stability. The interaction between CSs and CF promotes stable interfacial forces, contributing to the outstanding mechanical robustness, acid/alkali durability, and ultraviolet-visible color fastness of colorized CFs. However, there are some critical concerns that need to be addressed. Firstly, there is a lack of clear explanation regarding the coloration mechanism of randomly arranged CSs on the CF surface and the reaction mechanism from glucose to CSs. Additionally, the data analysis presented is inconsistent and does not sufficiently support the conclusions drawn. These issues raise concerns about the overall quality and validity of the study. Therefore, the current manuscript is unacceptable for publication in Nature Communications. Furthermore, there are several points that need improvement within the manuscript.

Reply: We appreciate the Reviewer for the professional and constructive comments, which help us to improve the quality of our manuscript. We have taken into account all the insightful suggestions of the Reviewer and revised the manuscript accordingly. According to the Reviewer's comments, the revised manuscript has been amended and improved from the following aspects

Firstly, according to the comments, we further explained the coloration mechanism, which was affected by the CSs average diameter, which was determined by the hue of color ². Moreover, the CSs diameter size was investigated by Gaussian fitting based on the CSs diameter distribution, and the linearity relationship between the average CSs diameter and characteristic wavelength of different colors has been constructed. Meanwhile, based on

some excellent papers and characterization in this manuscript, we emphasized the vital reaction parameter from glucose to CSs including reactant concentration, hydrothermal temperature, and time¹⁵. The high-resolution SEM images have been measured to explain the shape of the CSs, which demonstrated the hemispherical attaching to the CF surface.

Secondly, as mentioned by the Reviewer, which referred to the structure of the peacock feather, we have performed extensive surface and cross-section analyses on the back feather of the peacock using high-resolution SEM. The obvious nanoparticles and regular structure could be observed on the surface and cross-section of the back feather, respectively. Factually, in this manuscript, the attention was focused on the nanoparticle structure of the surface of the peacock back feather. According to the Reviewer's comments, this section has been amended as the structure of the colored CF fabric is mimicked the protrusive nanostructures of the peacock's back feathers surface. To further investigate the structural colors resulting from disordered structures, the angle-resolved reflective spectra were used to verify the isotropic optical properties, which exhibited a stable color with no obvious changes for reflection spectra of 30° to 150°. Furthermore, the reflection spectra and K/S value, which represented the color hue and color strength, have been measured to explain the changes in the structure of CSs after soaking and washing in an acetic acid solution under vibration for 120 minutes at a rate of 60 times per minute for color fastness testing, demonstrating the stable structure of carbon sphere on the carbon fiber surface.

Thirdly, indeed, as the Reviewer pointed out, with the hydrothermal temperature of 250 °C, pure CSs were difficult to obtain. Therefore, the elemental analysis of CSs for 10CC as the representative sample has been employed to characterize carbon elements, which indicated that the carbon content is 67.17 %. As investigated by Hu et al.¹⁶, a low-temperature HTC process was conducted up to 250 °C, employing several chemical transformation cascades,

consisting of the dehydration, condensation, polymerization, and aromatization¹⁷, and among the HTC process of glucose, 5-hydroxymethylfurfural (HMF) as the reaction-driving dehydration products were produced by hexose¹⁸, which consistent with the characterization of FTIR spectrum and solid-state carbon-13 MAS NMR spectra. For the HTC process of glucose, the formation process and the final material structures are rather complicated¹⁶, and a core-shell structure has been proposed to clarify the growth of the carbonaceous materials, which displayed a structure consisting of a hydrophobic core and a stabilizing hydrophilic shell with a large number of reactive oxygen functional groups, such as hydroxyl/phenolic, carbonyl, or carboxylic¹⁷. In our manuscript, the promising synthesis method of CSs was introduced to self-grow on the surface of carbon fiber.

Comment 1. In the introduction, the authors review structural colors derived from periodic photonic crystals and amorphous photonic structures. They also mention the brilliant colors observed in the back feather of a peacock, which is supposedly originating from randomly disordered amorphous nanostructures. However, according to the literature [Wang et al., 2016; Yoshioka & Kinoshita, 2002], the yellow feather (Figure 1(a)) possesses a clear periodic photonic crystalline structure along the cross-sectional direction under the surface of a barbule. Therefore, the analogy drawn to the structure of the peacock feather in this context is not well-founded. Moreover, the SEM photographs of the peacock feather in Figure 1(d) and Figure S6 show a smooth surface at different magnifications with 10 μ m and 100nm scale bars. Consequently, the randomly amorphous nanostructures claimed to exist on the peacock feather are not observed. To clarify the microscopic structure of the peacock feather, it would be beneficial to analyze the cross-sectional structure of a barbule. Additionally, it is important to note

that the structure of CSs on the CF surface is unnecessary to mimic the structure of the peacock feather, as they share no similarities.

Reply: We sincerely thank the Reviewer for the critical comments on our work, and for sharing these excellent papers. We also sincerely appreciate the Reviewer that you give valuable suggestions to help us to improve the quality of this manuscript.

As mentioned by the Reviewer, significant attentions should be given to the cross-sectional structure for the barbule of peacock feathers. Therefore, we have carefully read these representative references mentioned by the Reviewer, including Wang et al., “Photonic Crystals with an Eye Pattern Similar to Peacock Tail Feathers”, *Crystals*, 2016, 6, 99; and Yoshioka & Kinoshita, “Effect of Macroscopic Structure in Iridescent Color of the Peacock Feathers”, *Forma*, 2002, 17, 169–181.

Firstly, as mentioned in the paper (Wang et al., Photonic Crystals with an Eye Pattern Similar to Peacock Tail Feathers”, *Crystals*, 2016, 6, 99), a facile fabrication of photonic crystals with an eye pattern similar to peacock tail feathers have been demonstrated, whose eye pattern structures are grass-green in the middle of core, the structure color turns to yellow or orange around the green central part. The obvious eye pattern was derived from the crack, which formed by the competing effect of the shrinkage force of poly (styrene-methyl methacrylate-acrylic acid) (Poly (St-MMA-AA)) latex particles and resistance force of substrate toward the latex during the solvent evaporation process. Secondly, in the article (Yoshioka & Kinoshita, “Effect of Macroscopic Structure in Iridescent Color of the Peacock Feathers”, *Forma*, 2002, 17, 169–181), the transverse cross-section of barbules for a blue feather and the transverse cross-section of a barbule for a yellow feather was investigated, which displayed the periodically arrayed particles and lattice structure of the particles causing the iridescent peacock feathers. Indeed, the various structures were observed in different areas. These excellent papers have been

cited in the Introduction section of the revised manuscript.

Furthermore, according to the professional comments of the Reviewer, we have analyzed the cross-sectional structure for the barbule of peacock back feather using SEM images, which is shown below in Appendix Fig.18.

Appendix Fig.18 a SEM image of the cross-sectional structure for the barbule of a peacock back feather. b-c SEM images of the internal structure of the barbule. d-f SEM images of the surface and longitudinal structures of the barbule. g-h SEM images of the cross-sectional and surface structures of the barbule. i SEM image of the surface structure of the barbule.

As shown in Appendix Figs. 18a-c, the layered organizations are apparent. Among them, the contact between the protrusion and the empty site can be discovered, and the aligned vertical and periodicity repeatability between the perforation, the protrusion site, and the empty site describes the possible stacking. On the surface of the barbule of the peacock back feather, the protrusive nanostructure on the surface formed during the natural growth of the feather, which is shown in Appendix Figs. 18 d-i. Inspired by this protrusive nanostructure, the structure can be exploited to construct artificial structural

color through the *in-situ* growth method. Then, we shift to the design, fabrication, and application of artificial structural color generated by unitizing the inspiration of naturally growing structures.

Appendix Fig. 6 **a** Photograph of the peacock back feather. **b** The angle-resolved reflectance spectroscopy of the central area (green) for the peacock back feather. **c-h** SEM images of the barbule surface at the central area of the peacock back feather.

Meanwhile, a striking structural color has been created by living organisms on the central area of the peacock back feather, exhibiting the unique characteristic of angle-independent property (Appendix Figs. 6a, b). Additionally, some spherical nanoparticles could also be observed on the surface of the back peacock feather in the high-resolution SEM images (Appendix Figs. 6 c-h). Inspired by this nanostructure on the surface formed during the natural growth of the feather, a strategy of the randomly dynamic *in-situ* growth of bio-based CSs was proposed for the color construction of structural color on CF fabrics.

As shown in Appendix Fig. 18 and Appendix Fig. 6, according to the suggestions of the Reviewer, we have performed cross-sectional and surface

structures of the barbule located on the central area of peacock back feathers, and the angle-resolved reflective spectra of the central area of peacock back feathers were conducted. The characteristic wavelength of the angle-resolved reflective shows no obvious changes as the angle increases. The obvious nanostructure raises on the surface created by living organisms, most likely resulting from natural evolution, which inspires artificial structures by *in-situ* growth to construct structural color. Therefore, in the revised manuscript, we have amended the corresponding sentences, which are shown as follows.

“A striking structural color has been created by living organisms on the central area of the peacock back feather, exhibiting the unique characteristic of angle-independent property (Fig. 1a) (Appendix Fig. 6a, b in this file). Meanwhile, as shown in Fig. 1b and Supplementary Fig. 1 (Appendix Figs. 6c-h and Appendix Fig. 18 in this file), on the barbule surface for the central area of the feather, the protrusive nanostructures are formed during the natural growth of the feather. Inspired by this protrusive nanostructure, we proposed a strategy for the random *in-situ* growth of CSs on the CF fabric only using glucose as the feeding material (Fig. 1c, d). The CSs distributed and grew randomly by interacting with other CSs and CFs to form a structural color on the CF fabric, and the corresponding color tuning was designed by controlling glucose concentration during the one-step hydrothermal carbonization (HTC).”

The high-resolution SEM images of the Peacock back feather and SEM images of the surface of the colored CF fabric have been amended in **Fig.1**.

This part has been revised in the section of “**Results**”.

The extensive SEM images for the peacock’s back feathers have been added in the section of “**Supplementary information**”.

Thank you very much again for your professional comments and insightful suggestions, which helped us a lot to improve the quality of this manuscript.

Comment 2. The hydrothermal reaction of glucose with carbon fibers/fabrics raises several unanswered questions. The assumed reaction mechanism involves the hydrolysis and dehydration processes of glucose, followed by the production of polymeric intermediates. These intermediates are hypothesized to nucleate and grow on the surface of CFs, eventually carbonizing and forming the firmly attached CSs on the CF surface. However, there is a lack of evidence to support this proposed mechanism. Furthermore, it is unclear what the chemical composition of the carbon spheres is, especially considering the difficulty in obtaining pure carbon spheres under the hydrothermal temperature of only 250°C. Additionally, if the carbon spheres were to nucleate and grow on the CF surface, one would expect their shape to be mostly hemispherical. However, the SEM images in Figure 2(a)-(d) show a large number of spherical carbon spheres distributed and deposited on the CF surface, raising questions about the formation process.

Reply: We truly appreciate the respected Reviewer for the significant comments on our research work. We apologized for the incomplete expression of the reaction mechanisms. Therefore, according to the insightful comments of the Reviewer, we have conducted the examinations based on the FTIR spectrum, solid-state carbon-13 MAS NMR spectra, MALDI-TOF mass spectra, elemental analysis, TEM images, and analysis of some excellent papers. The possible reactions of glucose under the hydrothermal temperature of 250 °C for 2 h, as well as the structure and carbon contents of CSs have been proposed. Additionally, high-resolution SEM images were carried out to investigate the shape of CSs attaching to the surface of CF. We have amended this section in the revised manuscript, which is presented below.

Firstly, the synthesis of uniform carbonaceous particles from the glucose unitizing HTC process has been previously reported ¹⁹. However, in the HTC process of carbohydrates, the formation process and the final material

structures are rather complicated, which was mainly due to the formation of a multitude of furan-type dehydrated intermediates from carbohydrates, the complexity of the chemistry^{20, 21, 22}. As investigated by Hu et. al.¹⁶, a low-temperature HTC process was conducted up to 250 °C, employing several chemical transformation cascades, consisting of dehydration, condensation, polymerization, and aromatization¹⁷. During the hydrothermal treatment of glucose, 5-hydroxymethylfurfural (HMF) as the reaction-driving dehydration product is formed in the initial stage¹⁸, as well as formic acid as the degradation product^{23, 24}. The acidic condition promoted further dehydration of glucose into HMF, which subsequently occurred polymerization–polycondensation reactions leading to the formation of polyfuranic compounds^{25, 26}, as confirmed by solid-state carbon-13 MAS NMR and FTIR (Appendix Fig.19). During this polymerization step, the polymer phase separates from the aqueous solution to form the nucleation²⁷. A core-shell chemical structure has been proposed to clarify the growth of the carbonaceous materials, which displayed a structure consisting of a hydrophobic core and a stabilizing hydrophilic shell¹⁷. Depending on the reaction condition, the CSs can achieve various diameter sizes. When the temperature increases above 180 °C, the aromatic nature of CSs enhance via intramolecular condensation, dehydration, and decarboxylation reactions of polyfuranic chains. As demonstrated in Appendix Fig. 19b, the solid-state carbon-13 MAS NMR spectra were obtained from hydrothermally treated glucose at 250 °C at 2 h residence time. The sample was characterized by the presence of the central aromatic peak at 125–129 ppm, which clarified the tendency of aromatization. The final degree of aromatization of the CSs mainly dependent on the reaction temperature, since this parameter determines fraction of oxygen containing functional groups. For the HTC process of glucose, the possible reaction process is shown in Appendix Fig. 20. In our manuscript, the promising synthesis method of CSs on the CF was introduced to the construction of structural colors.

Appendix Fig. 19 a FTIR spectra of solids after the HTC process with various concentrations of glucose (XCC: X g 70 mL⁻¹). b Solid-state carbon-13 MAS NMR spectra of 13CC. c MALDI-TOF mass spectra of 13CC. d Supernatant solution separated from residue after the HTC process with various concentrations of glucose (SXCC: supernatant solution separated from residue of HTC process for X g 70 mL⁻¹).

Secondly, the emphasis is on elaborating the critical reaction conditions of HTC, which affected the fabrication of CSs. The reaction environment is an integrated system whereby influencing parameters have individual or synergetic effects, such as reaction temperature, time, and concentration²⁸. The reaction temperature offered two fundamental functions, such as providing heat for dissociation to accelerate polymerization and aromatization reactions²⁹. Meanwhile, the residence time is an important parameter that determines the carbonization extent achieved within a given temperature³⁰. The evolution of carbon related chemical structures in the HTC process was affected by the combined influences of temperature and residence time, which stayed the same during the reaction. Therefore, the elemental analysis of CSs for 13CC as the representative sample has been further employed to characterize carbon elements. Appendix Table 1 shows that the value of carbon content is 67.17 %,

with a temperature of 250 °C and reaction time of 2 h, giving rise to a carbon-rich solid called hydrochar^{26, 31}, which was named CSs in our manuscript.

Appendix Fig. 20 Proposed model for glucose HTC with 250 °C for 2 h. **a** The possible reaction including main steps of dehydration, polymerization, polycondensation, polyfuranic, and aromatization. **b** TEM images of CSs.

Appendix Table 1. Carbon content of hydrochars derived by glucose.

Feedstock	Sample	Temperature (°C)	Time(h)	Carbon (%)
Glucose	13CC	250	2	67.17

Finally, based on the professional comments of the Reviewer, the high-resolution SEM images of colored CF fabric were carried out to investigate the shape attaching to the surface of CF. It can be seen from Appendix Fig. 21, as mentioned by the Reviewer, the hemispherical contact surface was indeed displayed from the various areas of colored CF fabric, which contributed to the stability of the structural color.

Appendix Fig. 21 SEM images of lateral morphologies of CSs on the colored CFs.

We have added the corresponding discussions in the revised manuscript, which is shown below.

“A complicated chemical process is related to the HTC reactions^{32, 33}, primarily owing to a multitude of furan-type dehydrated intermediates^{20, 21, 22}. In the HTC process of glucose to form CSs during the construction of structural colors, the reaction temperature offers two fundamental functions, providing heat for dissociation to accelerate the polymerization and aromatization reactions²⁹. Meanwhile, the residence time determines the extent of carbonization at a given temperature³⁰. The evolution of the carbon-related chemical structures in the HTC process was affected by the combined influence of temperature and residence time, which maintained constant during the reactions. Supplementary Table 1 (Appendix Table 1 in this file) shows that the value of carbon content was 67.17% under the reaction temperature of 250 °C and time of 2 h, thus producing a carbon-rich CSs^{26, 31}.”

“A hypothetical mechanism for the formation path of CSs on the CF fabric is demonstrated in Fig. 2h. As reported by Hu et al.¹⁶, a low-temperature HTC process was conducted up to 250 °C, employing several chemical transformation cascades, including dehydration, condensation, polymerization, and aromatization¹⁷. During the hydrothermal treatment of glucose, 5-

hydroxymethylfurfural (HMF), a reaction-driven dehydration product, was formed in the initial stage¹⁸, with formic acid as the degradation product^{23, 24}. The acidic conditions promoted the further dehydration of glucose into HMF, which subsequently underwent polymerization-polycondensation reactions leading to the formation of polyfuranic compounds^{25, 26}. During this polymerization step, the polymer phase separates from the aqueous solution to form nucleation²⁷, and a core-shell chemical structure was proposed to clarify the growth of the carbonaceous materials¹⁷. As shown in Supplementary Fig. 8 (Appendix Fig. 20 b in this file), the TEM images demonstrate the core-shell structure of CSs obtained in this work. Subsequently, the aromatic nature of CSs was enhanced via the intramolecular condensation, dehydration, and decarboxylation reactions of the polyfuranic chains. As shown in Fig. 2f (Appendix Fig. 19b in this file), the presence of sp² hybridized carbons at 125–129 ppm clarified the aromatization of the polyfuranic compounds. A promising synthesis method that forms unique morphologies and structures of CSs was utilized to construct structural colors. The formation process of the CSs was consistent with the self-growth on the CF surface. In a hydrothermal environment, CSs tend to be spherical owing to energy minimization, with further growth for a stable structure³⁴.”

This part has been revised in the section of “**Results**”.

Appendix Fig. 19 is nominated as Figs. 2e-g and Supplementary Fig. 7c, Appendix Fig. 20b is nominated as Supplementary Fig. 8, and Appendix Fig. 21 is nominated as Supplementary Fig. 6, which have been added to the section of “**Results**” and “**Supplementary Information**”.

Thank you very much again for your professional and insightful comments, which have helped a lot to improve the quality of this manuscript!

Comment 3. The mechanism of structural coloration remains unclear but

is of great importance, especially considering its attribution to the Mie scattering resonance of CSs in this study. Structural colors resulting from disordered structures possess isotropic optical properties, which can be characterized using angle-resolved reflective spectra. Furthermore, structural colors arising from Mie scattering resonance are highly dependent on the size and composition of the CSs. However, the manuscript does not present quantitative control over the size and composition of CSs. The distribution of CSs on the CF surface is theoretically expected to have a determinative effect on the resulting structural colors.

Reply: We sincerely thank the Reviewer for the professional comments and insightful suggestions. To further investigate the isotropic optical properties, the average diameter of CSs, and composition of the CSs, the angle-resolved reflective spectra, the size distribution of CSs for different colors, FTIR spectrum, solid-state carbon-13 MAS NMR spectra, and MALDI-TOF mass spectra were employed.

Appendix Fig. 22 Angle-resolved reflectance spectroscopy of the colored CF fabric (7CC as the representative).

First, according to the professional suggestions of the Reviewer, angle-resolved

reflective spectra were investigated, which was used to confirm that the color is indeed insensitive to the viewing angles. As shown in Appendix Fig. 22, angle-resolved reflective spectra had no obvious changes for 7CC as the represented sample, which demonstrated the apparent angle-independent.

We have added this part in the revised manuscript, which was showed below.

“To further investigate the directional independence of colored CF fabrics, the angle-resolved reflection spectra were investigated. As shown in Fig. 3m (Appendix Fig. 22 in this file), the colored CFs fabrics exhibit the same reflection peaks when the fixed angle of the reflection source was adjusted from 30° to 150°, which illustrates an obvious angle-independent structural color. Therefore, the developed color construction process may potentially promote the stable color of CFs.”

The angle-resolved reflectance spectroscopy of colored CF fabric has been added in **Fig. 3**.

The discussion of angle-resolved reflective spectra has been added in the section of “**Results**”.

Secondly, the reaction environment is an integrated system whereby influencing parameters have individual or synergetic effects, such as reaction temperature, time, and concentration²⁸. In our manuscript, the concentration of glucose as the only variable was used to control the diameter distribution of CSs. The reaction temperature offered two fundamental functions, such as providing heat for dissociation to accelerate polymerization and aromatization reactions²⁹. Meanwhile, the residence time is an important parameter that determines the carbonization extent achieved within a given temperature³⁰. The evolution of carbon-related chemical structures in the HTC process was affected by the combined influences of temperature and residence time, which stayed the same during the reaction. During the hydrothermal treatment of glucose, 5-hydroxymethylfurfural (HMF) as the reaction-driving dehydration

product is formed in the initial stage¹⁸, and the acidic condition promoted further dehydration of glucose into HMF, which subsequently occurred polymerization–polycondensation reactions leading to the formation of polyfuranic compounds, and the aromatic nature of CSs enhance via intramolecular condensation, dehydration and decarboxylation reactions of polyfuranic chains^{25, 26}, as confirmed by solid-state carbon-13 MAS NMR, FTIR, and MALDI-TOF mass spectra (Appendix Fig. 19). During this polymerization step, the polymer phase separates from the aqueous solution to form the nucleation²⁷. A core-shell chemical structure has been proposed to clarify the growth of the carbonaceous materials, which displayed a structure consisting of a hydrophobic core and a stabilizing hydrophilic shell¹⁷.

Appendix Fig. 19 a FTIR spectra of solids after the HTC process with various concentrations of glucose (XCC: X g 70 mL⁻¹). b Solid-state carbon-13 MAS NMR spectra of 13CC. c MALDI-TOF mass spectra of 13CC. d Supernatant solution separated from residue after the HTC process with various concentrations of glucose (SXCC: supernatant solution separated from residue of HTC process for X g 70 mL⁻¹).

Appendix Fig. 7 CIE diagram showing the color shift of 4CC, 7CC, 10CC, 13CC, and 17CC for the diameter change with respective x–y coordinates of (0.2716, 0.3019), (0.3909, 0.3814), (0.3955, 0.3600), (0.3276, 0.3165), and (0.3436, 0.3732), respectively.

Finally, the distribution of five colors is shown in Appendix Fig.7. With changes in average CSs diameters, the colors are distributed around the white point. The thickness determines the color saturation and the size determines the color hue². Explanations of structural colors for the correlation between the diameters of particles and the hues of structural coloration have been proposed so far. As the main parameter, particle size can tune the scattering properties of materials in controlling the color. Magkiriadou et al.¹ used a quantitative scattering model to propose that color appearance is dominated by backscattering resonances of individual particles. Schertel et al.² obtain blue, green, red, and purple colors repeatably by varying the polystyrene colloidal scatterer size. Moreover, this research proved that the particle size determined the color hue, and with the variation of particle size, the color was changed, in which blue is only possible for the smallest particle sizes. Furthermore, Mao et al.³ proposed a system-disordered nanostructure with a material-independent optical response. For neglecting variation, the indiscernible structural color was generated by the substitution of different materials at the same particle size. Chu et al.⁴ proved that the hue could be tuned by controlling the size of the spheres in the

amorphous photonic structures. Additionally, as investigated by Cho et al.⁵, the green color corresponded to the largest particle size developing from the peak of the reflectance spectra, compared to red and blue colors from the dips of the reflectance spectra.

Appendix Fig. 3 The diameter distribution for CSs nanoparticles exhibiting different colors on CF fabric, which are, from **a** blue, **b** yellow, **c** orange-red, **d** purple, and **e** green. **f** measured characteristic wavelengths as a function of the average CSs diameters.

According to the professional suggestions of the Reviewer, the correlation between the hues of structural coloration and the diameters of CSs has been investigated in the revised manuscript. Indeed, the structural colors of CF fabric are also related to the diameters of CSs, which was clarified by the curve appearing in Appendix Fig. 3f obtained from experimental results. The diameters of CSs on the colored carbon fabric were measured by Image J software with a large number of CS, which derived from the different glucose concentrations of 4CC, 7CC, 10CC, 13CC, and 17CC at the same reaction condition. Then, the reflection spectra of 4CC, 7CC, 10CC, 13CC, and 17CC were employed to investigate the characteristic peak, which was shown in Appendix Fig. 4. Therefore, the characteristic wavelength was recorded at 400, 430, 470, 540, and 570 nm for 4CC (blue), 7CC (yellow), 10CC (orange-red), 13CC (purple), and 17CC (green), according to the peak(4CC and 17CC), and

dip (7CC, 10CC, and 13CC) of reflection spectra, which was consistent with the wavelength corresponding to the maximum K/S value (K/S value is the ratio of the absorption coefficient to the scattering coefficient for an opaque object, and a K/S value was used to indicate the color strength).

Appendix Fig. 4 **a** Reflection spectra, and **b** K/S curves of 4CC, 7CC, 10CC, 13CC, and 17CC.

We have added the investigation of the relationship between average particle size and characteristic wavelength in the section of discussion in our revised manuscript, which was shown as follows.

“Moreover, the characteristic wavelengths of colored CF fabrics in the reflectance spectra were recorded at 400, 430, 470, 540, and 570 nm, respectively, demonstrating for blue (400 nm), yellow (430 nm), orange-red (470 nm), purple (540 nm), and green (570 nm), according to the peaks (4CC and 17CC) and dips (7CC, 10CC, and 13CC) in Fig. 1g (Appendix Fig. 4a in this file). As shown in Supplementary Fig. 3 (Appendix Fig. 4b in this file), the K/S value at the characteristic wavelength was used to assess the color stability, and the characteristic peaks in K/S spectra correspond to the maximum K/S values at 700, 430, 470, 540, and 700 nm, matching with the dips of reflectance spectra. The diameter distribution and representative average diameter of CSs were estimated using the Gaussian function based on the diameter measurement of CSs on the colored CFs (Supplementary Fig. 4) (Appendix Figs. 3a–e in this file). The CSs with average diameters of 212.0, 258.7, 282.9,

308.6, and 350.0 nm on the CFs contributed to the blue, yellow, orange-red, purple, and green structural colors, respectively. The relationship curve between the characteristic wavelengths and CSs diameters for various colors is displayed in Fig. 1h (Appendix Fig. 3f in this file), further demonstrating that the average diameter of CSs on the CFs can tune the diversity of colors by controlling the concentration of the glucose solution.”

This part has been added in the section of “**Results**”.

Thank you very much for your valuable comments to help improve the quality of our manuscripts.

Comment 4. The physical implication and definition of the K/S value are unclear. While it is generally used for pigment dyeing and can be applied to characterize structural colors, its specific relevance to this study should be clarified. Additionally, it would be valuable to explain the changes in the structure of CSs after soaking and washing in an acetic acid solution under vibration for 120 minutes at a rate of 60 r/min for colorfastness testing.

Reply: We appreciate the Reviewer’s insightful suggestions and professional comments, which help us to improve the quality of this manuscript. Indeed, the physical implication and definition of the K/S value should be a clear explanation.

The definition of the K/S value is a function ($K/S=(1-R)^2/2R$) developed by Kubelka and Munk, who theorized in 1931 that the ratio of the coefficient of light absorption (K) to the coefficient of light scattering (S) is corresponding to the fractional reflectance of the light (R) of the opaque substrate at a given wavelength^{12, 13}. Meanwhile, the reflection spectra and K/S spectra, which demonstrated the hue and color strength of colored CF fabric, were investigated to explain the changes in the structure of CSs.

As suggested by the Reviewer, the definition of the K/S value was explained below.

“K/S value is a function ($K/S=(1-R)^2 / 2R$) developed by Kubelka and Munk, who theorized in 1931 that the ratio of the coefficient of light absorption (K) to the coefficient of light scattering (S) is corresponding to the fractional reflectance of the light (R) of the opaque substrate at a given wavelength^{12, 13}. The colored CF fabric K/S value could be directly measured by spectrophotometer at λ_{max} , which was presented with $K/S=(1-R_{\lambda_{max}})^2 / 2R_{\lambda_{max}}$. And the K/S value (Supplementary Fig. 3) (Appendix Fig. 23 in this file) at maximum absorption wavelength, corresponding to the characteristic wavelength at the peak or dip of reflection spectra, represented the color strength. Therefore, the wavelength corresponding to the maximum K/S value, matching with the characteristic wavelengths of reflectance spectra, represented the color strength used to assess the stability of color strength after different treatments.”

Appendix Fig. 23 K/S curves of 4CC, 7CC, 10CC, 13CC, and 17CC.

This part has been added in the section of “Supplementary Information” and “Results”.

According to the professional suggestion of the Reviewer, in the revised manuscript, the change of color was used to explain the changes in the structure of CSs after soaking and washing in an acetic acid solution (acetic

acid (CH_3COOH , 50 wt%)) water solution) under vibration for 120 minutes at a rate of 60 times per minute for colorfastness testing.

Therefore, the reflection spectrum and K/S spectra were employed to estimate the color hue and color strength. The reflection spectra and K/S spectra were measured to investigate the hue and color strength with 7CC colored CF fabric as the representative sample. As exhibited in Appendix Fig. 24, comparing to the raw 7CC colored CF fabric, the reflection spectra and K/S spectra, after soaking and washing in an acetic acid solution (acetic acid (CH_3COOH , 50 wt%)) water solution) under vibration for 120 minutes at a rate of 60 times per minute, indicated that color hue and strength showed no apparent changes. Appendix Fig. 25 showed the stable CSs structure and the hemispherical contact surface between CSs and C after soaking and washing in an acetic acid solution under vibration for 120 minutes at a rate of 60 times per minute. The stable structure of CSs is consistent with the stability of color and characteristic wavelength. To further investigated the stability of different colors, the K/S spectra after acetic acid solution washing at three different positions are demonstrated in Appendix Fig. 26. Comparing to the raw samples, the characteristic wavelengths and K/S value of color maintain stability, which indicates the stable CSs structure on the colored CF fabric.

Appendix Fig. 24 a Reflectance spectra of raw colored CF fabric, cleaning with an acetic acid solution. b and c the K/S spectra of raw colored CF fabric, washing and soaking with an acetic acid solution for 7CC at three different positions (7CC was selected as the representative sample).

Appendix Fig. 25 SEM images of 7CC as the represented sample after the soaking and washing in an acetic acid solution under vibration for 120 minutes at a rate of 60 times per minute.

Appendix Fig. 26 K/S spectra of soaking and washing of acetic acid solution test after mechanical rubbing test at 60 times per minute of vibration rate for 120 min **a** 4CC, **b** 7CC, **c** 10CC, **d** 13CC, and **e** 17CC, at three different positions.

This part has been amended in the section of “**Results**” and “**Supplementary Information**”.

Thank you very much for your constructive comments, which help us a lot to improve the quality of our manuscript.

Comment 5. Figure 5 illustrates the weaving process of the segment-color yarn to create colored CF fabric. However, it is not explained how the colorful yarn is continuously produced through the hydrothermal methodology in a batch process. Furthermore, there are concerns regarding the potential damage to the photonic structure of CSs on the

CF surface during the weaving process, which could eventually lead to the deterioration of the fabric colors. These aspects need to be addressed and discussed in more detail.

Reply: Thank you very much for the careful reading of our manuscript, and for arising this question. We apologized that the fabrication of the colorful yarn was ignored to explain in the manuscript. The fabrication of multi-colored CF yarns was the same process as colored CF fabric preparation. Therefore, in the revised manuscript, we have added this part in the section of “The preparation of the color carbon fiber fabric”. We agree with the Reviewer that the potential damage to the photonic structure of CSs on the carbon fiber surface during the weaving process could eventually lead to the deterioration of the fabric colors. It is indeed a great challenge of weaving multi-colored CF yarns, and we are planning to put more effort into finding a solution for this challenge in the future.

Appendix Fig. 9 Color stability after mechanical rubbing cycles of 0, 10, and 30 times. The measured reflectance spectra of colored CF fabrics for **a** 4CC, **b** 7CC, **c** 10CC, **d** 13CC, and **e** 17CC. **f** K/S values of these samples (Tx; T: mechanical rubbing test; x: the cycle times)

Indeed, attention to the mechanical stability of structural color should be attracted. According to the comments of the Reviewer, based on the ISO 105-X12, to indicate the mechanical stability of structural colors with 10 rubbing

cycles under a 50 kPa load pressure using a crock meter ¹¹. The durable carbon sphere structure and strong interactions between CSs and CFs afforded potential possibilities for the weaving process. To further demonstrate the color stability of the samples, we conducted a mechanical rubbing test at a load pressure of 50 kPa for 30 cycles, the reflectance spectra, K/S values, and 3D optical microscope images demonstrated that the colored CF fabrics exhibited satisfactory fastness to the mechanical rubbing of 30 cycles which was shown in Appendix Fig. 9 and Appendix Fig. 10. Meanwhile, the multi-colored CF yarns were prepared using the above-developed method to verify their applicability.

Appendix Fig. 10 3D optical microscope images after friction resistance test of cycling 0 (T0), 10 (T10), and 30 (T30) times for 4CC, 7CC, 10CC, 13CC, and 17CC.

We have revised this part in the section, which showed as below.

“The durable carbon sphere structure and strong interactions between CSs and CFs afforded potential possibilities for the weaving process. To further demonstrate the color stability of the samples, we conducted a mechanical rubbing test at a load pressure of 50 kPa for 30 cycles, the reflectance spectra (Supplementary Figs. 14a–e) (Appendix Figs. 9a-e in this file), K/S values (Supplementary Fig. 14f and Supplementary Table 4) (Appendix Figs. 9f in this file), and 3D optical microscope images (Supplementary Fig. 15) (Appendix Fig. 10 in this file) demonstrated that the colored CF fabrics exhibited satisfactory fastness to the mechanical rubbing of 30 cycles. Meanwhile, the multi-colored CF yarns were prepared using the above-developed method to verify their applicability.”

This part has been amended in the section of “**Results**”.

Appendix Fig. 9 and Appendix Fig.10 were nominated as Supplementary Fig. 14 and Supplementary Fig. 15, which have been added in the section of “**Supplementary Information**”.

Appendix Fig. 27 **a** Multi-colored CF yarns prepared using the one-step HTC of glucose. **b** Photograph of a multi-colored CF yarn. **c** Schematic illustration of the production process of colored CF fabric from multi-colored CF yarn. **d** Weaving process of preparing colored CF fabric. **e** Photograph of a package of multi-colored CF yarn. **f** Photograph of the weaving machine. **g** Photograph of the weaving process for the multi-colored CF yarn fabric. **h** A piece of 15×15 cm² plain weave fabric formed by industrial weaving machines.

“The one-step hydrothermal reaction was conducted by regulating the glucose construction, contributing to the *in-situ* self-growth of CSs on the surface of CF yarns (Fig. 5a) (Appendix Fig. 27a in this file). The multi-colored CF yarn produced by this method is shown in Fig. 5b (Appendix Fig. 27b in this file).

Therefore, the colored CFs were adapted to the varying mechanical requirements of industrial weaving machines. Figs. 5c-d (Appendix Figs. 27c-d in this file) shows the schematic illustration for the weaving fabrication of multi-colored CF fabric using the as-prepared multi-colored yarns. Here, we fabricated a plain multi-colored CF fabric on the weaving machine using the as-prepared CF yarns as both the warp and weft yarns (Figs. 5e-g) (Appendix Figs. 27e-g in this file). Although the CF yarns generally experienced severe mechanical rubbing with the heddles of the weaving machine during the weaving process, a piece of 15×15 cm² plain CF fabric was successfully produced and exhibited satisfactory multi-color (Fig. 5h) (Appendix Fig. 27h in this file). Owing to the following main reasons, the developed method may be potentially competitive in the industrial production of colored CFs. Firstly, glucose was used as the only feeding material in the color construction process, which is relatively low-cost and easily obtained. Secondly, glucose concentration was the only factor affecting the color variety under the constant reaction time and temperature. Therefore, we anticipate this developed approach can be potentially applied in the industrial color construction of CFs with multi-colors.”

Appendix Fig. 27 is nominated as Fig. 5, which have been added in the section of “**Results**”.

This part has been added in the section of “**Results**”.

Thank you very much for your professional and insightful comment to help improve the quality of our manuscripts.

Comment 6. The overall English expression in the manuscript requires significant improvement. Numerous errors and inaccurate sentences need to be corrected to enhance clarity and readability.

Reply: Thank you very much for the careful reading of our manuscript. We have carefully proofread the manuscript and presented a revised version of the

manuscript accordingly. According to the professional suggestions of Reviewer, we have checked the manuscript to avoid unclear words.

“These findings support the notion that CSs are a promising candidate for the structural color of CF fabrics, ...” has been corrected to “The process of color construction potentially promoted the color development of carbon materials.”

The word “lesion” has been corrected to “lesson”.

The spelling mistake of “titled” has been corrected to “tilted”.

The word “endows” should be corrected to “endowing”.

“to design” has been corrected to “for designing”.

“has emerged as” has been corrected to “have emerged as”.

The word “choice” has been amended to “choices”.

The word “renewable” has been amended to “renewability”.

The sentence “color fastness was evaluated by color fastness, acid pickling, and light fastness” has been corrected to “color fastness was evaluated by mechanical stability, acid pickling, and light fastness”.

“Inspired by the structure of brilliant peacock back feathers” has been amended to “inspired by the naturally grown protrusive nanostructures on the green central surface of peacock back feathers”.

“endows it with many outstanding properties of mechanical robustness” has been corrected to “endowing it with outstanding mechanical robustness”.

“inspired by the robust structural colors derived from back feather of peacock” has been amended to “inspired by the naturally grown protrusive nanostructures on the green central surface of peacock back feathers”.

“Striking structural coloration has been created by living organism” has been corrected to “A striking structural color has been created by living organisms on

the central area of the peacock back feather”.

“Inspired by the disordered structure of peacock’s back feathers” has been corrected to “inspired by the naturally grown protrusive nanostructures on the green central surface of peacock back feathers”.

“segment-color CF yarns were prepared using a one-step hydrothermal reaction of glucose.” has been corrected to “The one-step hydrothermal reaction was conducted by regulating the glucose construction, contributing to the in-situ self-growth of CSs on the surface of CF yarns.”

“The CSs can be randomly in-situ grown on the surface of CFs, contributing to the color tuning of CFs.” has been added in the section of “Discussion”.

We have carefully checked the revised Manuscript and Supplementary Information. The unclear and inaccurate words have been amended in the revised manuscript in red color.

Comment 7. By addressing these concerns and making the suggested improvements, the manuscript could be significantly enhanced in terms of scientific rigor, clarity, and overall quality of presentation.

Reply: We sincerely appreciate your professional and critical comments, which help us to improve the scientific rigor, clarity, and overall quality of this manuscript. According to your insightful suggestions, we have amended the manuscript.

Furthermore, we have carefully checked the revised manuscript and Supporting Information with these changes and point-by-point responses to the reviewer’s comments as appropriate. We appreciate for Reviewers' warm work earnestly and hope that these corrections will meet with approval.

Thank you very much again! Your kind consideration of this revised manuscript will be greatly appreciated.

Appendix Reference

1. Magkiriadou, S., Park, J.G., Kim, Y.S. & Manoharan, V.N. Absence of red structural color in photonic glasses, bird feathers, and certain beetles. *Phys. Rev. E Stat. Nonlin. Soft Matter Phys.* **90**, 062302 (2014).
2. Schertel, L. et al. The structural colors of photonic glasses. *Adv. Opt. Mater.* **7**, 1900442 (2019).
3. Mao, P. et al. Disorder - induced material - insensitive optical response in plasmonic nanostructures: vibrant structural colors from noble metals. *Adv. Mater.* **33**, 2007623 (2021).
4. Chu, J. et al. Edible amorphous structural color. *Adv. Opt. Mater.* **10**, 2102125 (2021).
5. Cho, S. et al. Selective coloration of melanin nanospheres through resonant Mie scattering. *Adv. Mater.* **29**, 1700256 (2017).
6. Narkevicius, A. et al. Revealing the structural coloration of self-assembled chitin nanocrystal films. *Adv. Mater.* **34**, e2203300 (2022).
7. Wang, Q. et al. Precise Regulation of Emission Maxima and Construction of Highly Efficient Electroluminescent Materials with High Color Purity. *Angew. Chem. Int. Ed.* **62**, e202301930 (2023).
8. Kinoshita, S., Yoshioka, S. & Miyazaki, J. Physics of structural colors. *Rep. Prog. Phys.* **71**, 076401 (2008).
9. Chang, Y. et al. Hereditary character of photonics structure in pachyrhynchus sarcitis weevils: color changes via one generation hybridization. *Adv. Opt. Mater.* **8**, 2000432 (2020).
10. Zhao, Y. et al. Scalable structural coloration of carbon nanotube fibers via a facile silica photonic crystal self-assembly strategy. *ACS Nano* **17**, 2893-2900 (2023).
11. Niu, W. et al. Multicolored photonic crystal carbon fiber yarns and fabrics with mechanical robustness for thermal management. *ACS Appl. Mater. Inter.* **11**, 32261-32268 (2019).
12. Kale, B.M. et al. Dyeing and stiffness characteristics of cellulose-coated cotton fabric. *Cellulose* **23**, 981-992 (2016).
13. Becerir, B. A novel approach for estimating the relation between K/S value and dye uptake in reactive dyeing of cotton fabrics. *Fibers Polym.* **6**, 224-228 (2005).
14. Wang, Z. et al. Structural evolution of hydrothermal carbon spheres induced by high temperatures and their electrical properties under compression. *Carbon* **121**, 426-433 (2017).
15. Zhao, Q., Tao, S., Miao, X. & Zhu, Y. A green, rapid, scalable and versatile hydrothermal strategy to fabricate monodisperse carbon spheres with tunable micrometer size and hierarchical porosity. *Chem. Eng. J.* **372**, 1164-1173 (2019).
16. Hu, B. et al. Engineering carbon materials from the hydrothermal carbonization process of biomass. *Adv. Mater.* **22**, 813-828 (2010).
17. Sevilla, M. & Fuertes, A.B. Chemical and structural properties of carbonaceous products obtained by hydrothermal carbonization of saccharides. *Chem. Eur. J.* **15**, 4195-4203 (2009).

18. Titirici, M.-M., Antonietti, M. & Baccile, N. Hydrothermal carbon from biomass: a comparison of the local structure from poly- to monosaccharides and pentoses/hexoses. *Green Chem.* **10**, 1204 (2008).
19. Sun, X. & Li, Y. Colloidal carbon spheres and their core/shell structures with noble-metal nanoparticles. *Angew. Chem. Int. Ed.* **43**, 597-601 (2004).
20. Kabyemela, B.M., Adschiri, T., Malaluan, R.M. & Arai, K. Glucose and fructose decomposition in subcritical and supercritical water: detailed reaction pathway, mechanisms, and kinetics. *Ind. Eng. Chem. Res.* **38**, 2888-2895 (1999).
21. Asghari, F.S. & Yoshida, H. Kinetics of the decomposition of fructose catalyzed by hydrochloric acid in subcritical water: formation of 5-Hydroxymethylfurfural, levulinic, and formic acids. *Ind. Eng. Chem. Res.* **46**, 7703-7710 (2007).
22. Baccile, N. et al. Structural characterization of hydrothermal carbon spheres by advanced solid-state MAS ¹³C NMR investigations. *J. Phys. Chem. C* **113**, 9644–9654 (2009).
23. Jin, F. et al. Hydrothermal conversion of carbohydrate biomass into formic acid at mild temperatures. *Green Chem.* **10**, 612 (2008).
24. Ischia, G. et al. Hydrothermal carbonization of glucose: Secondary char properties, reaction pathways, and kinetics. *Chem. Eng. J.* **449**, 137827 (2022).
25. Modugno, P. & Titirici, M.M. Influence of reaction conditions on hydrothermal carbonization of fructose. *ChemSusChem* **14**, 5271-5282 (2021).
26. Jung, D., Zimmermann, M. & Kruse, A. Hydrothermal carbonization of fructose: growth mechanism and kinetic model. *ACS Sustain. Chem. Eng.* **6**, 13877-13887 (2018).
27. Falco, C., Baccile, N. & Titirici, M.M. Morphological and structural differences between glucose, cellulose and lignocellulosic biomass derived hydrothermal carbons. *Green Chem.* **13**, 3273 (2011).
28. Zhuang, X. et al. A review on the utilization of industrial biowaste via hydrothermal carbonization. *Renew. Sustain. Energ. Rev.* **154**, 111877 (2022).
29. Nizamuddin, S. et al. An overview of effect of process parameters on hydrothermal carbonization of biomass. *Renew. Sust. Energ. Rev.* **73**, 1289-1299 (2017).
30. Funke, A. & Ziegler, F. Hydrothermal carbonization of biomass: A summary and discussion of chemical mechanisms for process engineering. *Biofuel. Bioprod. Biorefin.* **4**, 160-177 (2010).
31. Sevilla, M., Fuertes, A.B. & Mokaya, R. High density hydrogen storage in superactivated carbons from hydrothermally carbonized renewable organic materials. *Energy Environ. Sci.* **4**, 1400 (2011).
32. Cui, L.P. et al. Hydrolysis and carbonization mechanism of cotton fibers in subcritical water. *New Carbon Mater.* **33**, 245-251 (2018).
33. Ryu, J., Suh, Y.W., Suh, D.J. & Ahn, D.J. Hydrothermal preparation of carbon microspheres from mono-saccharides and phenolic compounds. *Carbon* **48**, 1990-1998 (2010).
34. Liu, S., Wang, X., Zhao, H. & Cai, W. Micro/nano-scaled carbon spheres based on hydrothermal carbonization of agarose. *Colloid. Surface. A* **484**, 386-393 (2015).

REVIEWER COMMENTS

Reviewer #1 (Remarks to the Author):

The authors have well addressed my concerns during the first-round review. I would like to recommend the acceptance of this manuscript by Nature Communications.

Reviewer #2 (Remarks to the Author):

In the revised manuscript, the authors have addressed my concerns and revised the manuscript. Many experimental results, such as the quantitative characterization of the color purity and directional independence, has been discussed in detail. I would like to suggest the acceptance of the revised manuscript for publication on Nat. Commun. after addressing the following minor problem.

In the reply to the 4th question of reviewer #2, why does the intensity of reflected spectrum of the colored CF fabrics increase with the number of mechanical rubbing cycles increasing?

Reviewer #3 (Remarks to the Author):

I have already read the author's response and the revised manuscript. The author has spent a significant amount of time and effort in addressing the reviewers' comments, correcting many errors, images, and textual content. This has greatly improved the readability and standardization of the manuscript compared to the version before the revisions. However, there are still several important issues that the author's response and the revised manuscript do not clearly address:

1. The mechanism of Carbon Sphere growth on carbon fibers using the Hydrothermal Carbonization (HTC) method is not clearly explained. It is unclear whether glucose molecules aggregate and then nucleate and grow on the fiber surface or whether glucose molecules decompose and then nucleate and grow after carbonization.
2. The mechanism for color generation is not well-explained. The intuitive feeling is that the disordered array of nanoscale carbon spheres scatters and resonates light (Mie Scattering). Unfortunately, the manuscript provides limited discussion on this vital part. Discussing the mechanism of color generation is crucial, and it should be explained through experimental and theoretical simulations to understand how changes in the size and structure of nanoscale carbon spheres affect the reflection spectra (colors).
3. The author claims that this is a scalable method, but there is no clear explanation or idea provided for the continuous production of fibers/fabrics.

These points are crucial for the novelty of this work, but unfortunately, the manuscript does not provide clear explanations and answers. This lack of clarity greatly affects both the novelty and readability of this work, making it appear to be merely reporting a method for preparing colored carbon fibers.

Therefore, I recommend that this work may not be suitable for publication in Nature Communications. The author may consider submitting it to other specialized journals.

Reviewer #1:

The authors have well addressed my concerns during the first-round review. I would like to recommend the acceptance of this manuscript by Nature Communications.

Reply: We thank the Reviewer for reviewing our manuscript and recommending the acceptance of our manuscript. We sincerely appreciate you for providing valuable suggestions and professional comments to improve the quality of our manuscript.

Thank you very much for the careful review of our work and the great contribution to the improvement of our manuscript!

Reviewer #2

In the revised manuscript, the authors have addressed my concerns and revised the manuscript. Many experimental results, such as the quantitative characterization of the color purity and directional independence, have been discussed in detail. I would like to suggest the acceptance of the revised manuscript for publication on Nat. Commun. after addressing the following minor problem.

Reply: We sincerely thank the Reviewer for the careful reading and positive comments on our manuscript. Meanwhile, we feel great thanks for your professional review work and insightful suggestions to improve our manuscript.

Comment 1: In the reply to the 4th question of reviewer #2, why does the intensity of the reflected spectrum of the colored CF fabrics increase with the number of mechanical rubbing cycles increasing?

Reply: We sincerely appreciate the valuable comments. We are sorry for the incomplete presentation in the reply to the 4th question of Reviewer #2 on the reflected spectrum. The reflected spectrum of the colored CF fabrics with different mechanical rubbing cycles was drawn by “Stacked Lines by Y Offsets” in the Origin software to clearly show the reflectance curves. The reflected curves are used to present the stability of the hue by the consistency of characteristic wavelength in these reflected spectra.

In the revised manuscript, we have explained that the wavelengths at characteristic peaks or dips of the reflected spectrum were used to present the hue for various colored CF fabrics. Meanwhile, the average K/S value at the characteristic peak in the K/S spectrum of various colored CF fabrics was used to evaluate the color depth after mechanical rubbing cycles. In the revised manuscript, we emphasized the corresponding details, which are shown below.

“Consistency of characteristic wavelengths of the reflectance spectra (Supplementary Figs. 19a–e), the stable K/S values (Supplementary Fig. 19f and Supplementary Table 4), and 3D optical microscope images (Supplementary Fig. 20) demonstrated that the colored CF fabrics exhibited satisfactory fastness to the mechanical rubbing of 30 cycles with stable color hue and depth”.

This part has been amended in the section of “**Textile weaving using multi-colored CF yarns**”

Thank you very much again for your valuable suggestions and acceptance of our manuscript!

Reviewer #3

I have already read the author's response and the revised manuscript. The author has spent a significant amount of time and effort in addressing the reviewers' comments, correcting many errors, images, and textual content. This has greatly improved the readability and standardization of the manuscript compared to the version before the revisions. However, there are still several important issues that the author's response and the revised manuscript do not clearly address.

Reply: Thank you very much for your recognition of our previous efforts on this manuscript! We appreciate the Reviewer for the insightful suggestions and professional comments, which helped a lot to improve the quality of this manuscript. According to the constructive comments of the Reviewer about the mechanism of carbon sphere growth and color generation, we have performed systematic characterization, analysis, and explanation to elucidate these two aspects.

Comment 1 The mechanism of Carbon Sphere growth on carbon fibers using the Hydrothermal Carbonization (HTC) method is not clearly explained. It is unclear whether glucose molecules aggregate and then nucleate and grow on the fiber surface or whether glucose molecules decompose and then nucleate and grow after carbonization.

Reply: We sincerely thank the Reviewer for the critical comments on our work. We also sincerely appreciate the Reviewer sharing these valuable suggestions to help us improve the quality of this manuscript. We have addressed your concerns with additional experiments and characterization.

To clearly explain the mechanisms of carbon spheres (CSs) growth on carbon fibers (CFs) using the hydrothermal carbonization (HTC) method, the photographs of the reaction products and the SEM images of CFs at various reaction times are further demonstrated. Meanwhile, X-ray photoelectron

spectroscopy (XPS) and Nuclear Magnetic Resonance (NMR) are used to investigate the transformation mechanisms of glucose into HTC carbon at different reaction stages. According to these experimental investigations on the mechanisms of CSs growth on CFs using the as-proposed HTC method, we have concluded that the growth of CSs on CFs could be divided into four main steps. The first step is 'Formation of furan compounds', the second step is 'Polymerization or condensation of furan compounds', the third step is 'Nucleation of aromatic clusters', and the fourth step is 'Formation of carbon spheres'. Therefore, we can answer that, as suggested by the Reviewer, during this HTC method, glucose molecules were firstly decomposed and then nucleated, and finally grew after carbonization. The detailed results and explanations are presented below.

According to the professional comments of the Reviewer, we have investigated the growth process of CSs and CFs by differing the reaction time at the same reaction conditions. The growth process of CSs mainly includes the formation of furan, polymerization or condensation of the intermolecular dehydration, nucleation with aromatization via intramolecular dehydration, and formation of CSs (Appendix Fig. 1a). Firstly, changes in the reaction process can be observed by the variation in reactant colors at different reaction times (Appendix Fig. 1b1-e1). The colors of reaction liquids were turned from yellow to brownish black with the increase of reaction time. As shown in Appendix Fig. 1b2-d2, SEM images of the CFs at various reaction times indicate that there are no obvious aggregates on the surface of CFs until 90 min. After 120 min of reaction (Appendix Fig. 1e2), a large number of CSs could be observed on the surface of CFs. These results suggest that there is no aggregation of particles on CFs during the initial stage of the reaction. Therefore, CSs were gradually formed on the surface of CFs.

Appendix Fig. 1 Photographs of reaction product at **a1** 30 min, **b1** 60 min, **c1** 90 min, and **d1** 120 min in the HTC reaction process. SEM images of carbon fibers (CFs) at different reaction times with **a2** 30 min, **b2** 60 min, **c2** 90 min, and **d2** 120 min in the HTC reaction process.

Secondly, to further explain the changes in products at the different stages of the hydrothermal reaction, X-ray photoelectron spectroscopy (XPS) is performed to investigate the functional groups of raw glucose and the solid products after hydrothermal reaction at different reaction times. As shown in Appendix Fig. 2, after the hydrothermal reaction, the surface C/O ratio of the solid significantly increases, which is significantly different from the raw sample (Appendix Fig. 2a). Compared to the raw glucose (50.93%), the surface carbon content of the solid product increased to 73.97% within 30 min of the reaction. In the subsequent reaction process, the carbon content is maintained in a range of 71.08%–75.33% at different reaction times. In the

C1s spectra, C–C/C=C, C–O, C=O, and O–C=O are assigned to 285.0, 286.3, 287.5, and 288.9 eV¹, respectively. Compared with the raw glucose, the significant change of the C1s XPS has been displayed with the forming of C=C, O–C=O, and C=O bonds via the hydrothermal reaction. The presence of these oxygenated groups was confirmed by the O1s spectrum. The signals of O1s spectra are attribute to the O-C (531.0 eV) and O=C (532.1 eV)², which also confirm the formation of oxygenated groups O=C bonds during the hydrothermal reaction³. The oxygen-containing functional groups present on the surface of solid products may consist of more reactive/ hydrophilic groups⁴. Combining with the photographs of products (Appendix Fig. 1b1-e1) and SEM images in Appendix Fig. 1b2-e2 at different reaction stages, XPS analysis in Appendix Fig. 2a-e shows the changes of functional groups for the glucose and solid products at the various reaction stages. The molecular structure of the reactant has changed markedly before the formation of obvious aggregations with the enhancement ratio of C/O and the formation of O–C=O, and C=O bonds, due to the dehydration and decomposition in the procedure of hydrothermal reaction⁵. Therefore, it can be concluded that the glucose molecules were first decomposed before the formation of aggregates.

Appendix Fig. 2 XPS survey spectra, C1s, and O1s spectra after hydrothermal reaction of glucose at 200 °C from different stages **a** glucose, **b** 30 min, **c** 60 min, **d** 90 min, and **e** 120 min at 200 °C.

Appendix Fig. 3 Solution ^{13}C NMR spectra of **a** 30 min and **b** 60 min after hydrothermal reaction of glucose at 200 °C. Solid ^{13}C NMR spectra of **c** 90 min and **d** 120 min after hydrothermal reaction of glucose at 200 °C.

Thirdly, Appendix Figs. 3a-d show the time evolution of the 110-150 ppm region during the hydrothermal reaction of glucose at 200 °C, which mainly demonstrates the aromatization of the polyfuranic compounds. In the early stages of the reaction, the region is characterized by the presence of the two peaks, which are due to the furanic rings (140–153 and 110–120 ppm). At 30 min, the products obtained from glucose of hydrothermal have a polymer-like structure composed of polyfuranic chains domains ⁶, along with the practical absence of a central peak at 125-129 ppm (Appendix Fig. 3a). As the HTC residence time increases, the relative intensity of the central peak at 125-129 ppm starts forming (Appendix Fig. 3b), and the production of solids (Appendix Fig. 1b). Corresponding to the SEM image of 30 min to 120 min, the NMR spectra show the relative intensity of the peaks at 140–153 ppm, and 110–120 ppm were assigned to the carbons of $\text{C}\alpha$ and $\text{C}\beta$ assigned to the furanic ring. With the extension of reaction time, the obvious intensity of the

peak at 125–129 ppm is observed, which can be assigned to carbon atoms belonging to aromatic rings, demonstrating the aromatization of the polyfuranic compounds. Compared to the solid ^{13}C NMR spectra of 90 min, the relative intensity of the peak at 125–129 ppm enhanced, demonstrating the increased degree of aromatization. As shown in Appendix Figs.1-3, due to the intramolecular dehydration, condensation, and decarboxylation, the more condensed sp^2 hybridized-aromatic chemical species was created ⁷. Nucleation then occurs at the critical supersaturation point of insoluble clusters. Furthermore, the active groups of the surface promote the growth of CSs with the enhancement of hydrothermal carbonization ^{4, 5}. Therefore, it can be concluded that the furan compounds were formed with the decomposition of glucose at the initial stage, and then furan compounds were polymerized or condensed, followed by the nucleation of aromatic clusters. As the reaction proceeded, carbon spheres were gradually formed and grew up on the surface of CFs.

In summary, the hydrothermal carbonization of glucose at different reaction stages has been conducted to investigate the formation of CSs on CFs. The glucose molecules decompose and then nucleate and grow after carbonization. The furan compounds were formed at the early stage, accompanied by polymerization or condensation of the intermolecular dehydration and fragmented products. The aromatization of soluble polymers is then produced via intramolecular dehydration, which causes the continuous formation of insoluble products promoting the nucleation of aromatic clusters. The formed nuclei grow with the reactive oxygen surface functionalities with the more condensed sp^2 hybridized-aromatic chemical species in the reaction process.

We have revised this part in the section of “**Formation of structural color by in-situ self-growth of CSs**” and “**Supplementary Information**”, which is shown below.

“The photographs of reaction products and SEM images with the corresponding reaction times are demonstrated in Supplementary Fig. 10, indicating that the CSs were formed gradually on the surface of CF with the hydrothermal reaction of glucose. To further explain the growth process of CSs, XPS and NMR were performed to investigate the products at the various stages of the hydrothermal reaction. Before the formation of aggregations on the surface CF, comparing to the raw glucose (Supplementary Fig. 11a), the functional group of the products has changed markedly with the enhancement ratio of C/O and the formation of O–C=O, and C=O bonds (Figs. 2f-i and Supplementary Figs. 11b-e), which is mainly due to the dehydration and decomposition in the process of hydrothermal reaction ⁵. Furthermore, in the early stages of the reaction (Supplementary Fig. 12a), the solution ¹³C NMR spectrum was characterized by the presence of the two peaks (140–153 and 110–120 ppm), which are assigned to the a polymer-like structure composed of polyfuranic chains domains ⁶. With the extension of reaction time, the obvious intensity of the peak at 125–129 ppm demonstrates the aromatization of the polyfuranic compounds (Supplementary Fig. 12b). Compared to the solid ¹³C NMR spectra of 90 min (Supplementary Fig. 12c), the relative intensity of the peak at 125–129 ppm enhanced (Supplementary Fig. 12d), demonstrating the increased degree of aromatization. The furan compounds were formed at the early stage, accompanied by polymerization or condensation of the intermolecular dehydration and fragmented products. The aromatization of soluble polymers was then produced via intramolecular dehydration, which caused the continuous formation of insoluble products promoting nucleation of aromatic clusters and growth. An illustration for the formation process of CSs on the CF fabric is demonstrated in Fig. 2j. The structural colors are constructed in the HTC process of glucose during the formation of CSs. The *in-situ* self-growth process of the CSs on the CF surface contribute to the constructing the structural color of CF fabric.”

“The CSs growth process is investigated by treating the same concentration of glucose with four reaction stages of 30, 60, 90, and 120 min at 200 °C. The colors of reaction liquids were turned from yellow to brownish black with the increase of reaction time (Supplementary Fig. 10 a1-d1). As shown in Supplementary Fig. 10 a2-d2, SEM images are used to investigate the effects of reaction time on the CSs formation process. At 120 min of hydrothermal reaction (Supplementary Fig. 10 d2), a large number of CSs could be observed on the surface of the CFs. These results suggest that there is no aggregation of particles during the initial stage of the reaction, and CSs form gradually on the surface of CF with the time extension in the process of hydrothermal reaction.”

“X-ray photoelectron spectroscopy (XPS) was performed to investigate the functional groups of raw glucose and the solid products after hydrothermal reaction with reaction times of 30, 60, 90, and 120 min at 200 °C. As shown in Supplementary Fig. 11, after the hydrothermal reaction, the surface C/O ratio of the solid significantly increased, which is different from the raw glucose (Supplementary Fig. 11 a). Compared to the raw glucose (50.93%), the surface carbon content of solid product increased to 73.97% within 30 min of the reaction. In the subsequent reaction process, the carbon content was maintained in a range of 71.08%–75.33% at different reaction times. In the C1s spectra, C–C/C=C, C–O, C=O, and O–C=O are assigned to 285.0, 286.3, 287.5, and 288.9 eV, respectively ¹. The significant changes of the C1s XPS has been displayed with the forming of C=C, O–C=O, and C=O bonds via the hydrothermal reaction, which is different from the raw glucose (Fig. 2 f-i and Supplementary Fig. 11). The presence of these oxygenated groups was confirmed by the O1s spectrum. The signals of O1s spectra attribute to the O–C (531.0 eV), and O=C (532.1 eV) ², which confirm the formation of oxygenated groups O=C bonds during the hydrothermal reaction ³. The oxygen-containing functional groups present on the surface of solid products

may consist of more reactive/ hydrophilic groups ⁴. Combining with the photographs of products (Supplementary Fig. 10 a1-d1) and SEM images in Supplementary Fig. 10 a2-d2 at different reaction stages, Supplementary Fig. 11 shows the changes of functional group for the products at the various reaction stages.”

“Supplementary Fig. 12 shows the 110–150 ppm region during the hydrothermal reaction of glucose at 200 °C with the extension of time. In the early stages of the reaction, the region is characterized by the presence of two peaks, which are due to the furanic rings (140–153 and 110–120 ppm). At 30 min, the products obtained from glucose of hydrothermal have a polymer-like structure composed of polyfuranic chains domains ⁶, along with the practical absence of a central peak at 125–129 ppm (Supplementary Fig. 12 a). As the HTC residence time increases, the relative intensity of the central peak at 125–129 ppm starts forming (Supplementary Fig. 12 b) with the production of solids (Supplementary Fig. 10 c1). Corresponding to the SEM images, the NMR spectra show the relative intensity of the peaks at 140–153 ppm, and 110–120 ppm are assigned to the carbons of C α and C β assigned to the furanic ring. With the extension of reaction time, the obvious intensity of the peak at 125–129 ppm is observed, which can be assigned to carbon atoms belonging to aromatic rings, demonstrating the aromatization of the polyfuranic compounds. Compared to the solid ¹³C NMR spectra of 90 min, the relative intensity of the peak at 125–129 ppm enhanced in Supplementary Fig. 12 d, demonstrating the increased degree of aromatization. As shown in Supplementary Figs. 10-12, due to the intramolecular dehydration, condensation, and decarboxylation, the more condensed sp² hybridized-aromatic chemical species was created ⁷. Nucleation then occurs at the critical supersaturation point of insoluble clusters. Furthermore, the active groups of the surface promote the growth and formation of CSs with the enhancement of hydrothermal carbonization ^{4, 5}.”

Appendix Fig. 1 has been nominated as **Fig. 2h** and **Supplementary Fig. 10** in the revised manuscript.

Appendix Fig. 2 and **Appendix Fig. 3** have been nominated as **Supplementary Fig. 11** and **Supplementary Fig. 12**, which have been added in the section of “**Supplementary Information**”.

This part has been added in the section of “**Formation of structural color by in-situ self-growth of CSs**” and “**Supplementary Information**”.

Thank you very much for your professional comment on the mechanism of CSs growth, which helps us a lot to promote the growth mechanism in our manuscript!

Comment 2 The mechanism for color generation is not well-explained. The intuitive feeling is that the disordered array of nanoscale carbon spheres scatters and resonates light (Mie Scattering). Unfortunately, the manuscript provides limited discussion on this vital part. Discussing the mechanism of color generation is crucial, and it should be explained through experimental and theoretical simulations to understand how changes in the size and structure of nanoscale carbon spheres affect the reflection spectra (colors).

Reply: We are grateful to the Reviewer for the professional and critical comments on the insufficient discussion of the color generation mechanism in our previous version. We sincerely agree that a distinct explanation of the mechanism for color production is very important. Therefore, as suggested by the Reviewer, we have performed theoretical calculations to verify the nature of a single CS. Moreover, we have also conducted theoretical simulations of the random distribution of CSs with various diameters, which are based on the Mie scattering theory to understand the mechanism for color generation. In this part, the refractive index (n) and extinction coefficient (k) were performed

by the Ellipsometer (Ellipsometer, J. A. Woollam M-2000). The simulated reflectance spectra were calculated with MATLAB software (MATLAB, 2018) and commercial finite-difference time-domain software (FDTD, Lumerical).

Appendix Fig. 4 **a** Four random SEM images as an example to calculate coverage (Average coverage ratio (C_1) of 35.71%). **b** Complex refractive index of CSs (Complex refractive index $\tilde{n} = n_r + ki$). **c** Experimental Absorption (A_e) and **d** Reflectance (R_e) spectra with average diameters of 258.7, 282.9, and 308.6 nm, respectively. **e** Calculated spectra of absorption (A_c) and **f** backscattering (S_b) corresponding to the experimental average diameters.

The scattering ability of nanospheres was directly proportional to their refractive index. The electric and magnetic resonances based on Mie scattering in the visible region can be supported by the suitable diameter of a single dielectric sphere with an appropriate refractive index⁸. Chakrabarty et al.⁹ reported the complex refractive index of the spherical carbonaceous particles. Meanwhile, owing to the near-spherical shape of the particles, the Mie theory has been justified. As shown in SEM images of Appendix Fig. 1, the structural color, formed by randomly distributed *in-situ* growth CSs on the CF surface of various diameters, indicates that the color is dependent on the Mie scattering of individual nanospheres. To investigate the influence of different glucose concentrations on the coverage ratio of CSs, four random

SEM images were recorded as an example to calculate the coverage (Appendix Fig. 4a). The complex refractive index (\tilde{n}) of the CSs was measured by the Ellipsometer, which was fitted with Cauchy's equations for both real (n_r) and imaginary (k) coefficients to reflect wavelength dependence, which is shown in Appendix Fig. 4b.

Based on Mätzler's MATLAB codes of Mie scattering ¹⁰, we have calculated the absorption (A_c) and backscattering (S_b) spectra corresponding to the experimental average diameters, which are shown in Appendix Equation 1–12. Appendix Fig. 4c shows the absorption (A_e) spectra in the visible range with average CSs diameters of 258.7, 282.9, and 308.6 nm. The absorption monotonically decreases with the wavelength for different CSs diameters. The reflectance (R_e) spectra corresponding to the A_e spectra are demonstrated in Appendix Fig. 4d. The A_c spectra, calculated based on Mie scattering theory, are consistent with experimental absorption spectra, displaying monotonically decreases with the wavelength. The S_b spectra significantly depend on the wavelength, which indicates the same characteristic wavelength as the experimental reflectance spectra.

The calculated absorption (A_c) and backscattering (S_b) are defined as

$$A_c = \frac{C_A}{\pi r^2} \quad (1)$$

$$C_A = \sum_{n=1}^{\infty} (2n + 1) \{Re(a_n + b_n) - (|a_n|^2 + |b_n|^2)\} \quad (2)$$

$$S_b = \frac{C_S}{\pi r^2} \quad (3)$$

$$C_S = \frac{1}{2} |\sum_{n=1}^{\infty} (2n + 1) (-1)^n (a_n - b_n)|^2 \quad (4)$$

where C_A is the cross-section of Mie absorption, C_S is the cross-section of Mie backscattering, r is the radius of the spheres, and a_n and b_n are obtained by spherical Bessel functions ¹¹.

$$a_n = \frac{m^2 j_n(mx)[x j_n(x)]' - \mu_1 j_n(x)[mx j_n(mx)]'}{m^2 j_n(mx)[x h_n^{(1)}(x)]' - \mu_1 h_n^{(1)}(x)[mx j_n(mx)]'} \quad (5)$$

$$b_n = \frac{c_1 j_n(mx)[x j_n(x)]' - j_n(x)[mx j_n(mx)]'}{c_1 j_n(mx)[x h_n^{(1)}(x)]' - h_n^{(1)}(x)[mx j_n(mx)]'} \quad (6)$$

$$h_1^{(1)}(Z) = j_n(Z) + iy_n(Z)j_n(Z) \quad (7)$$

$$[Zj_n(Z)]' = Zj_{n-1}(Z) - nj_n(Z) \quad (8)$$

$$[Zh_1^{(1)}(Z)]' = Zh_{n-1}^{(1)}(Z) - nh_n^{(1)}(Z) \quad (9)$$

$$n_{max} = x + 4x^{\frac{1}{3}} + 2 \quad (10)$$

$$A_c = \frac{2}{x^2} \sum_{n=1}^{\infty} (2n+1) \text{Re}(a_n + b_n) - \frac{2}{x^2} \sum_{n=1}^{\infty} (2n+1) (|a_n|^2 + |b_n|^2) \quad (11)$$

$$S_b = \frac{1}{x^2} |\sum_{n=1}^{\infty} (2n+1) (-1)^n (a_n - b_n)|^2 \quad (12)$$

Where n_r is the refractive index of the sphere, $x=2\pi r/\lambda$, C_1 is the coverage ratio, $Z=x$ or $Z=mx$, and n ($n = 1, 2, 3$) based on spherical Bessel functions of order are described by Bohren and Huffman ¹².

The CSs absorb a broad band of visible light regardless of their size. However, the CSs on the surface of CF have wavelength-dependent backscattering since they display reflection color. The absorption suppresses the multiple scattering, providing the resonant colors predominant, which strongly depends on the average diameter. The characteristic wavelengths show colors complementary to the dips in the spectra, as shown in Appendix Figs. 4d, f. The mismatch between the calculated and the experimental spectra is attributed to the inaccurate measurement of the refractive index, the difference between single-particle and multi-particle arrangement, and the distinction of backscattering and reflectance. Nevertheless, the characteristic

wavelengths of backscattering spectra vary with the average diameter of CSs, which are consistent with dip or peak in the experimental reflectance spectra.

Appendix Fig. 5 Simulations for different colors using FDTD. **a** Schematic illustration of FDTD simulations for CSs with different average diameters. **b-f** FDTD-Simulated reflection spectra based on 212.0, 258.7, 282.9, 308.6, and 350.5 nm average diameter with a random distribution of different CSs diameter. **g** Diameter dependence of dip or peak position in reflectance and FDTD simulations reflectance spectra. **h** and **i** FDTD-simulated reflection spectra of 282.9 nm average diameter CSs at different chaos.

As shown in Appendix Fig. 5, FDTD-simulation was utilized to discuss the mechanism of color generation with random distribution of different CSs diameters. Mie scattering characteristics reflectance spectra based on the average diameters of 212.0, 258.7, 282.9, 308.6, and 350.5 nm are simulated to disclose different structural colors generated by Mie scattering (Appendix Fig. 5a). Furthermore, FDTD-simulated reflection spectra of 282.9 nm

average diameter with various spheres diameter at different chaos was used to investigate the influence of random distribution on color. The refractive index of the CSs is shown in Appendix Fig. 4b, which is vary with the wavelengths. To get closer to the random distribution in the experiment, we simulate the reflectance spectra with the random chaos for multi-spheres. The negative extinction coefficient does not violate energy conservation ¹³, implying that the CSs provide extra absorption. As shown in Appendix Figs. 5b-f, chaotic spheres within the range of diameter distribution, which represent the CSs on the surface for different colored CF fabrics, support the possibility of achieving consistent reflection spectra. More importantly, the color varies with the average diameter of CSs, which is contributed by Mie scattering from the individual nanospheres. We confirm the mechanism with good accordance between experimental reflectance spectra of the sphere random distribution and scattering spectra calculated by Mie theory. Appendix Fig. 5g indicates the characteristic wavelengths of the dip or peak in the stimulated scattering spectra. The slight mismatch between the simulated and the experimental spectra is attributed to the inaccurate measurement of the refractive index and the average diameter, which cannot fully representation of sphere distribution. Nevertheless, the characteristic wavelengths of dip or peak for different colored CF fabrics in the scattering spectra vary with the average diameter of CSs, which are consistent with dip or peak in the experimental reflectance spectra. Furthermore, based on the Mie scattering, we simulated the characteristics spectra with an average diameter of 282.9 nm, which was conducted under different chaos. The characteristic wavelengths remain unchanged with the different chaos of the spheres, as shown in Appendix Figs. 5h, i. The simulation indicates that the color generation of the colored CF fabrics accords with the Mie scattering theory. The color hue is dependent on the average diameter of the CSs on the surface of CFs.

In summary, according to the theoretical calculations of single sphere backscattering spectra based on the experimental average diameter and extended theoretical simulations of random distribution for different diameters, the color varies with the average diameter of CSs, which is contributed by Mie scattering from the individual spheres. The simulation results with good accordance with the experimental reflectance spectra of the sphere random distribution. The color generation of the colored CF fabrics accord with the Mie scattering theory, and the characteristic wavelength representing the hue of color is dependent on the average diameter of the CSs on the surface of CFs.

“As illustrated in Fig. 3, FDTD simulation is utilized to discuss the mechanism of color generation with a random distribution of different CSs diameters. Mie scattering characteristics reflection spectra based on average diameters of 212.0, 258.7, 282.9, 308.6, and 350.5 nm are simulated to disclose different structural colors (Fig. 3a). Furthermore, FDTD-simulated reflection spectra of 282.9 nm average diameter with various spheres diameter at different chaos is used to investigate the influence of random distribution on color. The refractive index of the CSs is shown in Supplementary Fig. 13, which varies with the wavelengths. To get closer to the random distribution in the experiment, we simulate the reflectance spectra with the random chaos for multi-spheres. As shown in Figs. 3b-f, chaotic spheres within the range of diameter distribution, which represent the CSs on the surface of different colored CF fabrics, support the possibility of achieving consistent reflection spectra. We confirm the mechanism with good accordance between experimental reflectance spectra of the sphere random distribution and simulated reflectance spectra calculated by Mie theory. More importantly, the color varies with the average diameter of CSs, which is contributed by Mie scattering from the individual nanospheres. Fig. 3g indicates the characteristic wavelengths of the dip or peak in the stimulated reflectance spectra. The

slight mismatch between the simulated and the experimental spectra is attributed to the inaccurate measurement of the refractive index and the average diameter, which cannot fully representation of sphere distribution. Nevertheless, the characteristic wavelengths of dip or peak in the simulated reflectance spectra vary with the average diameter, which is consistent with the dip or peak in the experimental reflectance spectra. Furthermore, based on the Mie scattering, we simulated the characteristics spectra with an average diameter of 282.9 nm, which was conducted under different chaos. Compared to the simulated reflectance spectrum in Fig. 3d, under the same average diameter, the diverse random distributions of different diameters show the same characteristic wavelength representing the hue of color. The characteristic wavelengths remain unchanged with the different chaos of the spheres, as shown in Figs. 3h, i. The simulation indicates that the color generation of the colored CF fabrics accord with the Mie scattering theory, and the color hue is dependent on the average diameter of the CSs on the surface of CFs.”

Appendix Fig. 5 has been nominated as **Fig. 3**, which has been added in the revised manuscript.

This part has been added in the section of “**Formation of structural color by in-situ self-growth of CSs**” in the revised manuscript.

We sincerely appreciate the respective Reviewer for professional comments! The insightful suggestions make a great contribution to help us improve the quality of our manuscript.

Comment 3 The author claims that this is a scalable method, but there is no clear explanation or idea provided for the continuous production of fibers/fabrics.

Reply: We thank the Reviewer for careful reading and pointing out this undefined expression of “scalable”. In this manuscript, we reported that various colored CF fabrics could be fabricated based on the in-situ growth of CSs on the CF fabric using glucose as the feeding material. Moreover, we also reported that the multi-colored CF yarns could be prepared using the above-developed method, and the multi-colored CF fabric could be weaved using the as-prepared CF yarns as both the warp and weft yarns on the weaving machine. Therefore, we hope that this proposed method may be extended to the appropriate reaction vessel using the prepared colored CFs with different sizes.

However, to avoid ambiguity in the expression of “scalable”, according to the constructive advice from the Editor, we have deleted the expression of “scalable” and “large-scale” in the revised manuscript. We hope that this amendment can meet with your approval.

We are sorry again for the undefined expression about the definition of scalable in our manuscript. Thank you very much for reviewing our manuscript. We really appreciate your constructive comments and careful work, which are valuable in improving the quality of our manuscript!

Appendix Characterization

Solid-state MAS NMR ^{13}C spectra were acquired on a Solid Nuclear Magnetic Resonance (NMR, Agilent DD2-500MHz). Solution-state NMR ^{13}C spectra were acquired on a Solution Nuclear Magnetic Resonance (NMR, Bruker Avance III HD 500MHz). The XPS spectra were performed by X-ray photoelectron spectroscopy with Al K α X-ray (XPS, ESCALAB 250Xi). The complex refractive index was performed by the Ellipsometer (Ellipsometer, J. A. Woollam M-2000). The calculation was acquired from MATLAB software (MATLAB, 2018). The simulated reflectance spectra were calculated with commercial finite-difference time-domain software (FDTD, Lumerical).

Appendix Reference

1. Yu, S. et al. Decoupled temperature and pressure hydrothermal synthesis of carbon sub-micron spheres from cellulose. *Nat. Commun.* 13, 3616 (2022).
2. Wang, F. et al. Selective adsorption–deposition of gold nanoparticles onto monodispersed hydrothermal carbon spherules: a reduction–deposition coupled mechanism. *J. Mater. Chem. A* 3, 1666 (2015).
3. Ryu, J., Suh, Y.W., Suh, D.J. & Ahn, D.J. Hydrothermal preparation of carbon microspheres from mono-saccharides and phenolic compounds. *Carbon* 48, 1990-1998 (2010).
4. Sevilla, M. & Fuertes, A.B. Chemical and structural properties of carbonaceous products obtained by hydrothermal carbonization of saccharides. *Chem. Eur. J.* 15, 4195-4203 (2009).
5. Sevilla, M. & Fuertes, A.B. The production of carbon materials by hydrothermal carbonization of cellulose. *Carbon* 47, 2281-2289 (2009).
6. Falco, C., Baccile, N. & Titirici, M.M. Morphological and structural differences between glucose, cellulose and lignocellulosic biomass derived hydrothermal carbons. *Green Chem.* 13, 3273 (2011).
7. Baccile, N. et al. Structural characterization of hydrothermal carbon spheres by advanced solid-state MAS ^{13}C NMR Investigations. *J. Phys. Chem. C* 113, 9644–9654 (2009).
8. Ren, J. et al. Mie resonant structural colors based on ZnO spheres and their application in multi-color pattern: Especially realization of red color. *Chem. Eng. J.* 474, 145530 (2023).
9. Chakrabarty, R.K. et al. Brown carbon in tar balls from smoldering biomass combustion. *Atmos. Chem. Phys.* 10, 6363-6370 (2010).
10. Cho, S. et al. Selective coloration of melanin nanospheres through resonant Mie scattering. *Adv. Mater.* 29, 1700256 (2017).
11. Mätzler, C. Matlab functions for Mie scattering and absorption. *IAP Res. Rep.* 8, 1 (2002).
12. Bohren, C.F. & Huffman, D.R. Absorption and scattering of light by small particles. *Wiley, New York* (1983).
13. Chen, Y.J., Lee, C.C., Chen, S.H. & Flory, F. Extra high reflection coating with negative extinction coefficient. *Opt. Lett.* 38, 3377-3379 (2013).

REVIEWERS' COMMENTS

Reviewer #1 (Remarks to the Author):

I am pleased to read the most recent version of the document and the author's response to the reviewer's comments. In the revised manuscript, the authors provide additional experiments, simulations, and analyses to clarify the mechanism of carbon sphere growth on carbon fibers and the mechanism for color generation. In their response, I believe that the authors have well addressed the concerns proposed by the reviewers.

In particular, I believe the following points have been well addressed:

(1) The additional experiments and analysis at different reaction times are convincing, indicating that the mechanism of carbon sphere growth on carbon fibers might include the glucose molecules decomposing and then nucleating and growing after carbonization. The recent experiments and characterization results can now sufficiently provide a much better understanding of the carbon sphere growth on fibers. The mechanism of carbon sphere growth now has been explained more deeply. With the improvement of the current work compared to the previous ones, I think that the authors have fully addressed the reviewer's earlier concerns.

(2) The revised manuscript has addressed the reviewer's concerns about the mechanism for color generation. The theoretical calculation explains the reviewer's confusion on scatters and resonates light (Mie Scattering) of nanoscale carbon spheres. I think the added FDTD simulations to investigate the effect of Mie scattering with a large enough disordered structure array of nanoscale carbon spheres, which is quite convincing to explain that the reflection spectra (colors) are dependent on the average size of carbon spheres. The simulation experiments of different colors are clear and serve to convincingly support the color generation mechanism.

(3) The revised version of the manuscript has been modified appropriately. The unclear words in this article have been changed to better mirror the message of the paper. The authors have made significant efforts to clarify the reviewer's concerns.

Based on the above reasons, I recommend the acceptance of this manuscript with no further concerns and revisions.

[Note from the Editor: Reviewer #1 was asked to look also over the response given to Reviewer #3]

After carefully reading the reviewer #3's comments, I do not think the concerns proposed by reviewer #3 are critical questions to this work. I believe that the authors have well addressed the concerns proposed by the three reviewers. Therefore, I recommend the acceptance of this manuscript by Nature Communications.

Reviewer #1:

I am pleased to read the most recent version of the document and the author's response to the reviewer's comments. In the revised manuscript, the authors provide additional experiments, simulations, and analyses to clarify the mechanism of carbon sphere growth on carbon fibers and the mechanism for color generation. In their response, I believe that the authors have well addressed the concerns proposed by the reviewers.

In particular, I believe the following points have been well addressed:

(1) The additional experiments and analysis at different reaction times are convincing, indicating that the mechanism of carbon sphere growth on carbon fibers might include the glucose molecules decomposing and then nucleating and growing after carbonization. The recent experiments and characterization results can now sufficiently provide a much better understanding of the carbon sphere growth on fibers. The mechanism of carbon sphere growth now has been explained more deeply. With the improvement of the current work compared to the previous ones, I think that the authors have fully addressed the reviewer's earlier concerns.

(2) The revised manuscript has addressed the reviewer's concerns about the mechanism for color generation. The theoretical calculation explains the reviewer's confusion on scatters and resonates light (Mie Scattering) of nanoscale carbon spheres. I think the added FDTD simulations to investigate the effect of Mie scattering with a large enough disordered structure array of nanoscale carbon spheres, which is quite convincing to explain that the reflection spectra (colors) are dependent on the average size of carbon spheres. The simulation

experiments of different colors are clear and serve to convincingly support the color generation mechanism.

(3) The revised version of the manuscript has been modified appropriately. The unclear words in this article have been changed to better mirror the message of the paper. The authors have made significant efforts to clarify the reviewer's concerns.

Based on the above reasons, I recommend the acceptance of this manuscript with no further concerns and revisions.

[Note from the Editor: Reviewer #1 was asked to look also over the response given to Reviewer #3]

After carefully reading the reviewer #3's comments, I do not think the concerns proposed by reviewer #3 are critical questions to this work. I believe that the authors have well addressed the concerns proposed by the three reviewers. Therefore, I recommend the acceptance of this manuscript by Nature Communications.

Reply: We sincerely thank the Reviewer for reviewing our manuscript and recommending the acceptance of our manuscript.

Thank you very much for the careful review of our work and the great contribution to the improvement of our manuscript!

Reviewer #2

(Remarks to the Author)

In the revised manuscript, the authors have deeply elucidated the mechanism of carbon sphere growth and the mechanism of color generation through supplementary experimental characterizations (X-ray Photoelectron Spectroscopy, Nuclear Magnetic Resonance), theoretical calculations and simulations (MATLAB and commercial finite-difference time-domain software). In detail, the authors have concluded that the growth of CSs on CFs can be divided into four main steps, which is derived from the formation of furan compounds, polymerization or condensation of furan compounds, nucleation of aromatic clusters and formation of carbon spheres, indicating that the mechanism of carbon sphere growth on carbon fibers include the glucose molecules decomposing and then nucleating and growing after carbonization. Moreover, the authors have fully explained the color generation mechanism, which accords with the Mie scattering theory and is dependent on the average size of carbon spheres, dispelling the reviewer's concerns. Thus, I would like to suggest the acceptance of the revised manuscript for publication on Nat. Commun. without further revisions.